# The mark of vegetation change on Earth's surface energy balance

Gregory Duveiller ⓘ [1], Josh Hooker[1] & Alessandro Cescatti[1]

Changing vegetation cover alters the radiative and non-radiative properties of the surface. The result of competing biophysical processes on Earth's surface energy balance varies spatially and seasonally, and can lead to warming or cooling depending on the specific vegetation change and background climate. Here we provide the first data-driven assessment of the potential effect on the full surface energy balance of multiple vegetation transitions at global scale. For this purpose we developed a novel methodology that is optimized to disentangle the effect of mixed vegetation cover on the surface climate. We show that perturbations in the surface energy balance generated by vegetation change from 2000 to 2015 have led to an average increase of $0.23 \pm 0.03\,°C$ in local surface temperature where those vegetation changes occurred. Vegetation transitions behind this warming effect mainly relate to agricultural expansion in the tropics, where surface brightening and consequent reduction of net radiation does not counter-balance the increase in temperature associated with reduction in transpiration. This assessment will help the evaluation of land-based climate change mitigation plans.

[1] Bio-Economy Unit, Directorate D - Sustainable Resources, European Commission Joint Research Centre, Via Enrico Fermi 2749, I-21027 Ispra, VA, Italy. Correspondence and requests for materials should be addressed to G.D. (email: gregory.duveiller@ec.europa.eu)

Terrestrial biomes are a large yet variable sink of $CO_2$[1] that play an important role in climate change mitigation[2]. Changes in vegetation cover alter the strength of this sink and may even turn it into a source of greenhouse gases, as a result of land processes such as deforestation[3]. In addition, changes in vegetation cover also imply changes in both radiative and non-radiative biophysical properties that may in turn affect the local climate and the surface energy balance[4–8]. As an example of the complexity of such biophysical land-climate interactions, the conversion of forests into grasslands typically entails a rapid increase in albedo[9] and a concomitant decrease in evapotranspiration that may ultimately lead to cooling or warming, depending on which of these two processes dominates[10,11].

To date the accounting of biophysical perturbations of the surface energy budget in major climate assessments has been mostly limited to changes in albedo[2]. However, there is an increasing body of literature that demonstrates how all the terms of the energy balance significantly impact the surface climate and should be accounted for in land-based mitigation plans[12–18]. The complex issue of land-climate biophysical interactions has been intensively explored with model-based studies[8,12,19,20] but the capacity of Earth system models (ESM) to represent accurately these biophysical properties and, in particular, the partitioning of available energy into latent and sensible heat fluxes, is still uncertain[19,21]. As a result, the assessment of extensive land cover changes performed with ESMs at continental to global scales[8,12,22] has led to contrasting predictions[15,19] and the impacts of smaller-scale changes in vegetation cover have only recently been evaluated[23]. What's more, assessments based on ground observations, such as those from flux sites and meteorological stations[7,24–26] typically have insufficient spatial coverage to derive conclusions at the global scale.

To cope with the uncertainty of model simulations and the limitations of surface data, satellite observations are increasingly used to produce data-driven diagnostics at large scales[9–11,27,28] because they combine global coverage with the high resolution required to assess local changes in vegetation cover. Until now, the terrestrial biophysical effect on climate has been derived from space observations with two alternative approaches. The first identifies the signal of the actual change in the local climate following an observed land cover transition[10,28,29], and is therefore limited both by the accuracy of land cover change detection from the satellite imagery, and by the frequency and extent of those changes during the observation period. The second, and more common, approach is based on a space-for-time approximation that identifies the potential impact of a land cover transition from differences in climate amongst neighbouring areas with similar environmental conditions but contrasting vegetation[9,11,27]. Although studies performed using either of these two approaches present a range of diagnostics of varying complexity and scope, none provide at the same time an assessment of the full energy balance, at global scale, and for multiple vegetation transitions, as now required for the comprehensive evaluation of land-based mitigation plans.

Here we present such an assessment. Our novel approach adopts the space-for-time logic to multi-scale remote sensing products to quantify the potential effect that a complete transition from one vegetation class to another would have on the individual components of the surface energy balance and on the resultant change in land surface temperature. This information is spatially and temporally explicit, enabling us to draw a comprehensive picture of the geographic and seasonal patterns of these potential changes. The resulting data set is freely available and fully described in an accompanying data description publication[30]. We use this global data set to quantify the total effect on the surface energy balance resulting from all vegetation changes that have occurred during the period 2000–2015, and then translate this effect into a change of $0.23 \pm 0.03\,°C$ over the concerned land. Agricultural expansion is most responsible for this increase, due to a decrease in evapotranspiration that is not compensated by an increase in albedo. We further show how all potential transitions towards croplands or grasslands raise local temperatures irrespective of the vegetation originally present. Similarly, converting tropical evergreen forests to any other vegetation cover results in a warming of the local climate.

## Results

**Changes in surface energy balance due to deforestation.** We start by analyzing the potential changes in annual radiometric land surface temperature (LST) following a broad vegetation transition from forest to crops or grasses. These are based on land cover information for 2010 and monthly satellite observations collected during the period 2008 to 2012. Figure 1 shows spatial patterns that are in line with previous studies focusing strictly on deforestation[10,29]. We identify a general increase in clear sky daytime LST particularly evident in regions where vegetation growth is typically limited by water availability, but also in regions where vegetation growth is limited by energy, such as Southeast Asia (Fig. 1a). In parallel, night-time temperatures consistently decrease across the mid-latitudes, while the tropics and areas beyond the polar circle show a mixture of mild decreases and increases (Fig. 1b). The mean of day-time and night-time LST can provide a rough estimate for air temperature. Mean LST follows a clear latitudinal/temperature gradient from a decrease in boreal zones to an increase in temperate to tropical areas (Fig. 1c), in line with studies based on in situ observations[7]. Finally, the diurnal amplitude in LST increases across the globe with the exception of some areas in North America and Russia (Fig. 1d).

In contrast to other global studies, here the underlying physical mechanisms behind temperature changes can be appreciated by looking at the variations in the components of the surface energy balance. Figure 2 shows that the forest to crops/grasses transition induces a general increase of reflected solar shortwave radiation (SW), as herbaceous canopies are typically brighter than trees[4,31]. This effect is particularly large at more northern latitudes, where trees can partly mask the highly reflective snow cover of the understory. It is worth noting that compensation of the reduction in absorbed SW in the energy balance equation depends on the background climate and therefore varies geographically, and is dominated by the reduction in sensible (H) and ground heat fluxes (G) in cold and/or humid climates at northern latitudes (Fig. 2d), as well as by the decrease in latent heat flux (LE) in warm and/or arid regions (Fig. 2c). These perturbations in the surface energy balance ultimately produce an increase in longwave (LW) upwelling radiation (and therefore of LST) that is particularly relevant in water-limited regions like southern Europe and western USA (Fig. 1a).

**Biophysical effects of multiple vegetation cover transitions.** The main novelty of this assessment with respect to previous work is the possibility to delve into finer levels of land cover conversion. Figure 3 summarizes the global annual perturbations of the surface energy balance expected for transitions among the following vegetation types: evergreen broadleaf forests (EBF), deciduous broadleaf forests (DBF), evergreen needleleaf forests (ENF), savannas (SAV), shrublands (SHR), grasslands (GRA), croplands (CRO) and wetlands (WET). The seasonal patterns for both the changes in surface energy balance and the resulting temperature are also provided respectively in Supplementary Figs. 1 and 2.

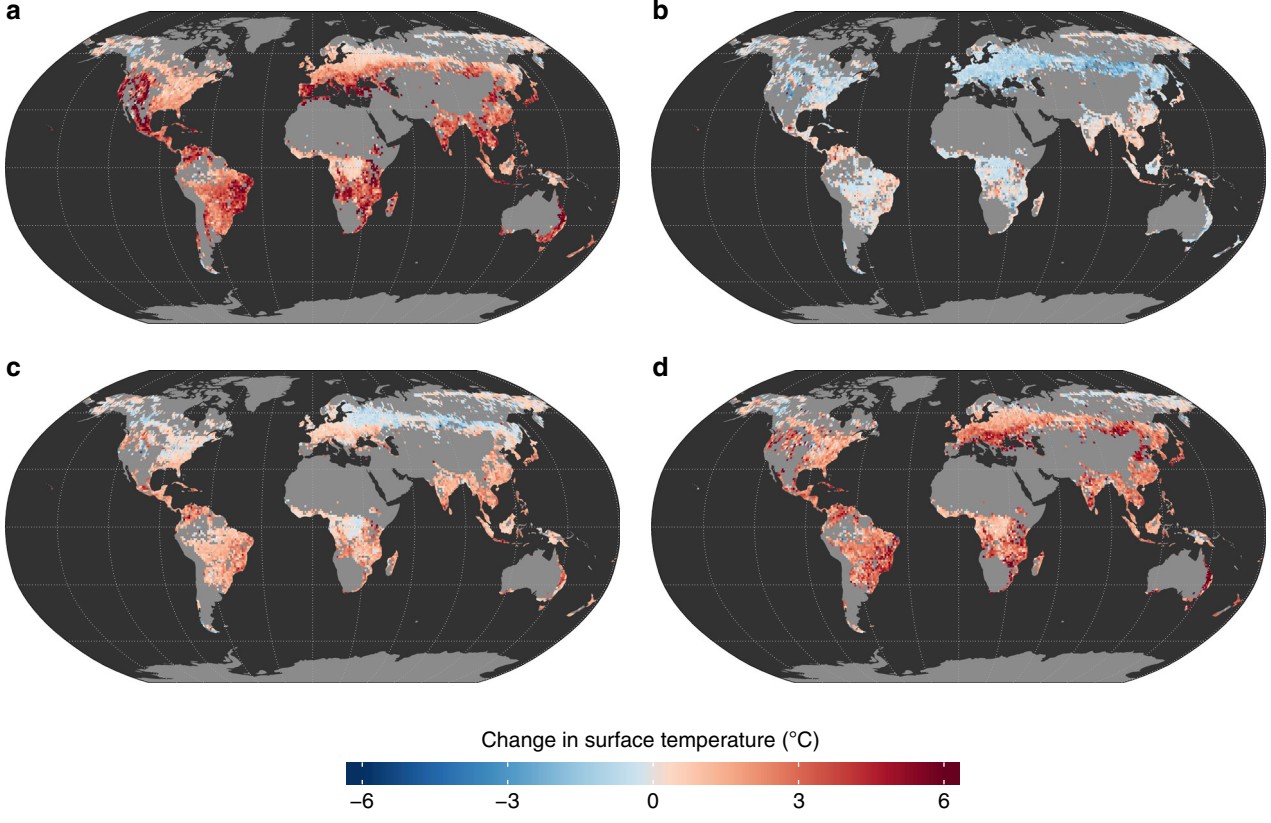

**Fig. 1** Potential changes to surface temperatures caused by deforestation. Panels describe the expected average annual change of **a** day-time and **b** night-time clear sky land surface temperature (LST), of **c** mean LST (defined as the average between **a** and **b**) and of **d** LST diurnal amplitude (defined as the difference between **a** and **b**)

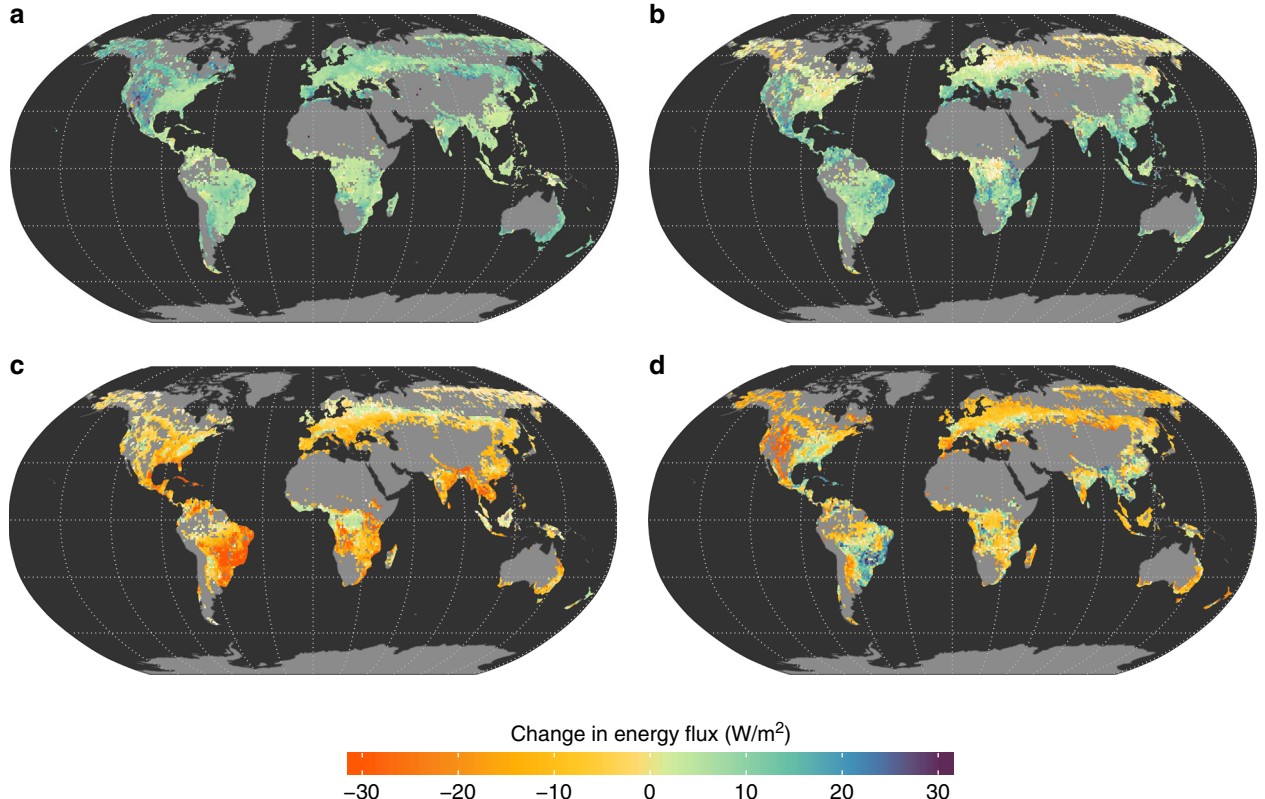

**Fig. 2** Potential changes to the local energy balance caused by deforestation. Expected average annual changes are provided for **a** shortwave reflected radiation (SW), **b** longwave emitted radiation (LW), **c** latent heat flux (LE) and **d** the combination of sensible and ground heat fluxes (H+G). Deforestation is considered here to be a conversion of forests to either crops or grasses

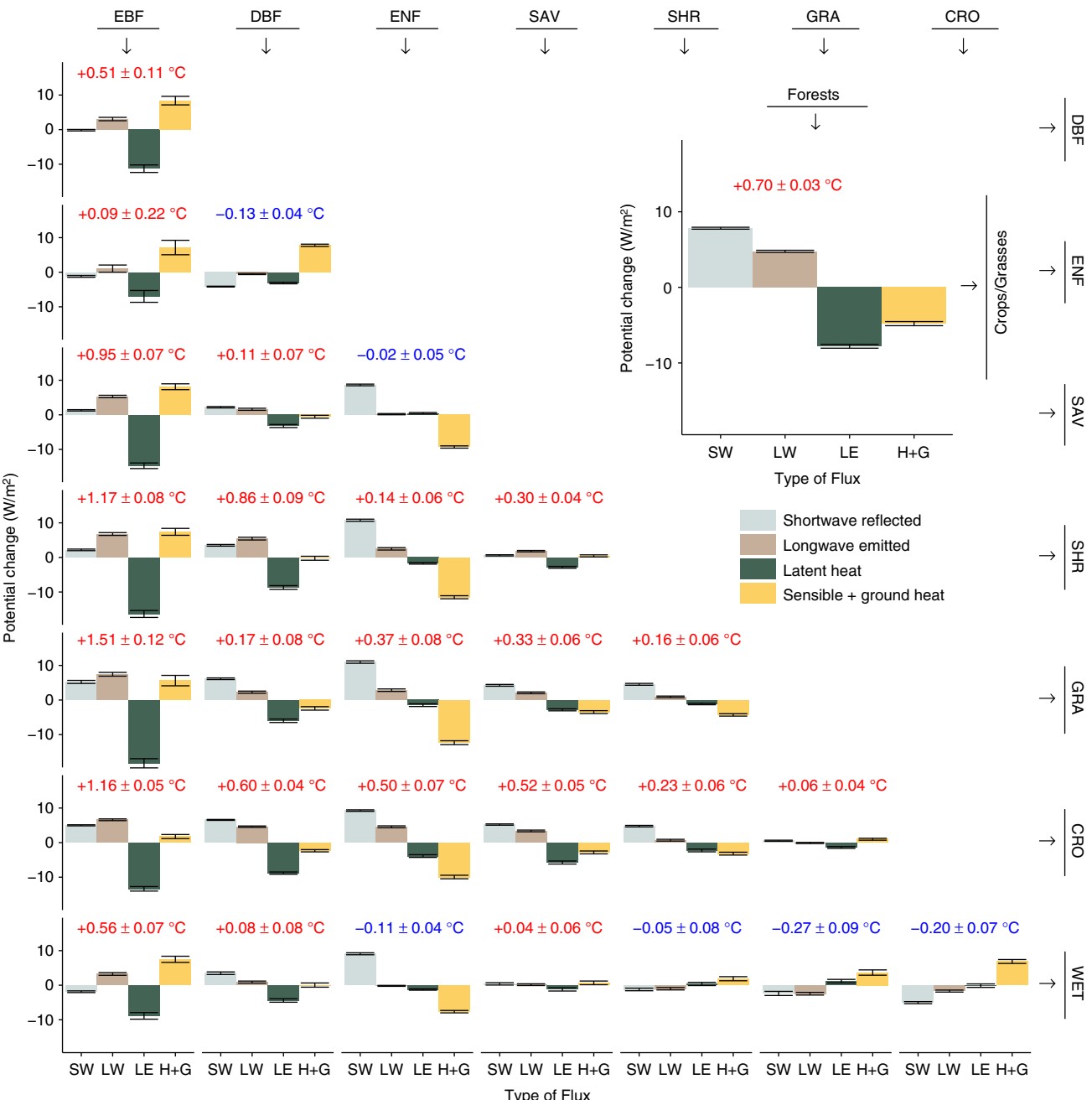

**Fig. 3** Global summary of the mean annual potential change in surface energy balance and temperature for various transitions in vegetation type as derived from satellite observations. The transitions shown involve the following vegetation classes: evergreen broadleaf forests (EBF), deciduous broadleaf forests (DBF), evergreen needleleaf forests (ENF), savannas (SAV), shrublands (SHR), grasslands (GRA), croplands (CRO) and wetlands (WET). Because transitions are symmetric, reverse transitions can be derived by inverting the sign. The inset shows a more generic transition from forests to either crops or grasses corresponding to the maps shown in Figs. 1 and 2. For each transitions, the mean change is provided for the shortwave reflected radiative flux (SW), longwave emitted radiative flux (LW), latent heat flux (LE) and the combination of sensible and ground heat fluxes (H+G). The number above the bars represents the mean surface temperature change observed for that transition ± two times the standard error around the mean, as do the confidence intervals represented on the bar charts of the flux values

Figure 3 illustrates how both the type of forest and the region where they are located influence how deforestation perturbs the surface energy balance. Loss in evergreen broadleaf forest (essentially confined to the tropics) resulting in a strong reduction in latent heat flux, whereas loss in evergreen needleleaf forest (located mostly in boreal zones) causes sensible and ground heat fluxes to decrease; deciduous forests have an intermediate response. While deforestation systematically results in higher radiative fluxes leaving the surface, the balance

between shortwave reflected and longwave emitted radiation changes depends on forest type. Deforestation of needleleaf trees show a stronger increase in reflected radiation, partly because these ecosystems are predominantly located in the Northern hemisphere, characterized by extended snow cover periods, but also because needleleaf trees are typically darker than their broadleaf counterparts. The vegetation type that replaces the forest also has an effect: for example, the reduction in latent heat flux is stronger when tropical forests are converted to grasslands

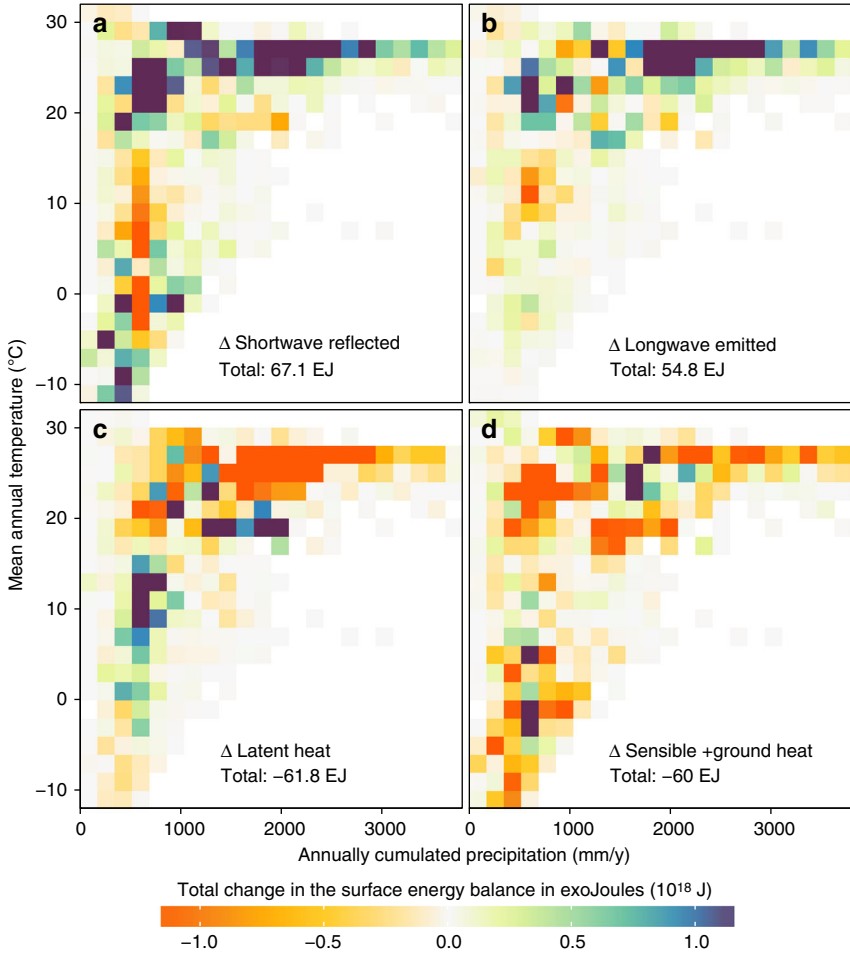

**Fig. 4** Effect of actual changes in vegetation cover from 2000 to 2015 on the surface energy balance. Each panel illustrates how this energy change in exoJoules (EJ) varies across climatic gradients of mean annual temperature and annually cumulated precipitation for the various components of the surface energy balance: **a** shortwave reflected radiation, **b** longwave emitted radiation, **c** latent heat, and **d** the combination of sensible and ground heat fluxes. Each panel also provides the corresponding total net effect. The climate axes are calculated based on CRU data v4.00 at 0.5° × 0.5° resolution

than when converted to croplands, suggesting that croplands have more access to water in the analyzed areas, possibly because of irrigation.

Beyond deforestation, other vegetation cover transitions that can also be explored include changes in species compositions, such as that from deciduous to needleleaf trees (DBF to ENF) which shows a strong increase in H+G, or changes from grasslands to croplands. Analyzing changes from forests to savannas, and from forests to shrublands, can provide information on the effect of tree density and tree height respectively. Finally, a striking pattern that emerges from Fig. 3 is that both the column covering tropical evergreen forests (EBF) and the rows representing agricultural expansion (CRO and GRA) consistently show warming, irrespective of which transition is considered. The driving forces behind this warming is a reduction in evapotranspiration in the former and increase in albedo in the latter.

**The biophysical effect of past vegetation cover changes**. The potential changes summarized in Figs. 1, 2 and 3 are used to estimate the global effects that recent changes in actual vegetation have had on the surface energy balance. For this purpose, the observed vegetation fraction change from 2000 to 2015 (based on respective maps from the ESA CCI project[32]) are multiplied by the corresponding changes in surface energy fluxes retrieved for each transition. The global perturbation resulting from all

transitions leads to a reduction in the surface energy budget of 121 exoJoules ($121 \times 10^{18}$ J). To put this number in perspective, this is almost one quarter of the total world supply of primary energy in 2015[33]. Exploring these cumulated changes across climatic gradients (Fig. 4) reveals a general brightening of the surface in the warm humid climates that is compensated by a strong reduction in latent heat, while mildly colder and drier climates show the opposite response (the changed area in each bin is shown in Supplementary Fig. 3). Very-cold climates show brightening (probably due to stronger snow albedo following forest cover reduction) counter-balanced by an increase in sensible heat flux. These perturbations of the surface energy balance are all summarized together at the global scale in Fig. 5, along with a subdivision according to specific vegetation transitions (for a similar plot with transitions ordered by magnitude of change, see Supplementary Fig. 4). Agricultural expansion into evergreen broadleaf forests has had an overwhelming effect on this perturbation, but conversion of deciduous broadleaf forests and shrublands into cropland also rank high in Fig. 5, consistently reducing the amount of energy available for evapotranspiration whilst brightening the surface. Changes from cropland back to forest also rank in the more important transitions, but they do not compensate the effects of deforestation.

Through a decomposition of the energy balance terms (see methods section), we can estimate how the changes reported in Fig. 5 translate into a change of surface temperature over the area

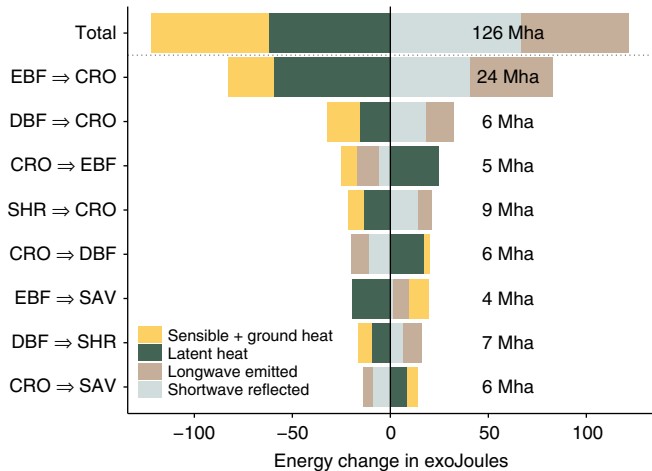

**Fig. 5** Cumulated changes in energy for each component of the surface energy balance resulting from recent major vegetation transitions. Transitions are sorted according to decreasing absolute change in the surface energy balance. The changed area per transition, calculated based on the ESA CCI land cover maps of 2015 and 2000, are reported in megahectares on the right of the bars. The transitions shown involve the following vegetation classes: evergreen broadleaf forests (EBF), deciduous broadleaf forests (DBF), savannas (SAV), shrublands (SHR) and croplands (CRO)

that underwent vegetation change from 2000 to 2015 (Fig. 6). Furthermore, we can highlight the role of each component in causing this change in temperature. At the global level, vegetation change has caused an average increase of 0.23 ± 0.03 °C where that vegetation change has occurred, driven by a warming effect from turbulent fluxes that is not compensated by the cooling effect caused by higher albedo. This increase in temperature follows a latitudinal gradient with the larger effect in the tropics, where the role of evapotranspiration is dominant, while at higher latitudes the warming is negligible and the brightening effect of vegetation change is counter-balanced only by sensible heat. The selected transitions in Fig. 6 confirm that conversions to croplands unambiguously increase the land surface temperature. However, it also shows how afforestation in the tropical belt has had a considerable cooling effect, though one that is not compensatory to that of deforestation because these changes have not occurred in the same regions. In boreal regions, the conversion of dense evergreen needleleaf forest to savannas or shrublands leads to a substantial increase in sensible heat instead of latent heat, counteracting the brightening effect that a reduction in canopy cover has on masking the snow.

## Discussion

This study makes the first global scale data-driven assessment of how different vegetation changes can influence the surface energy balance. Altogether, our results quantify these influences across

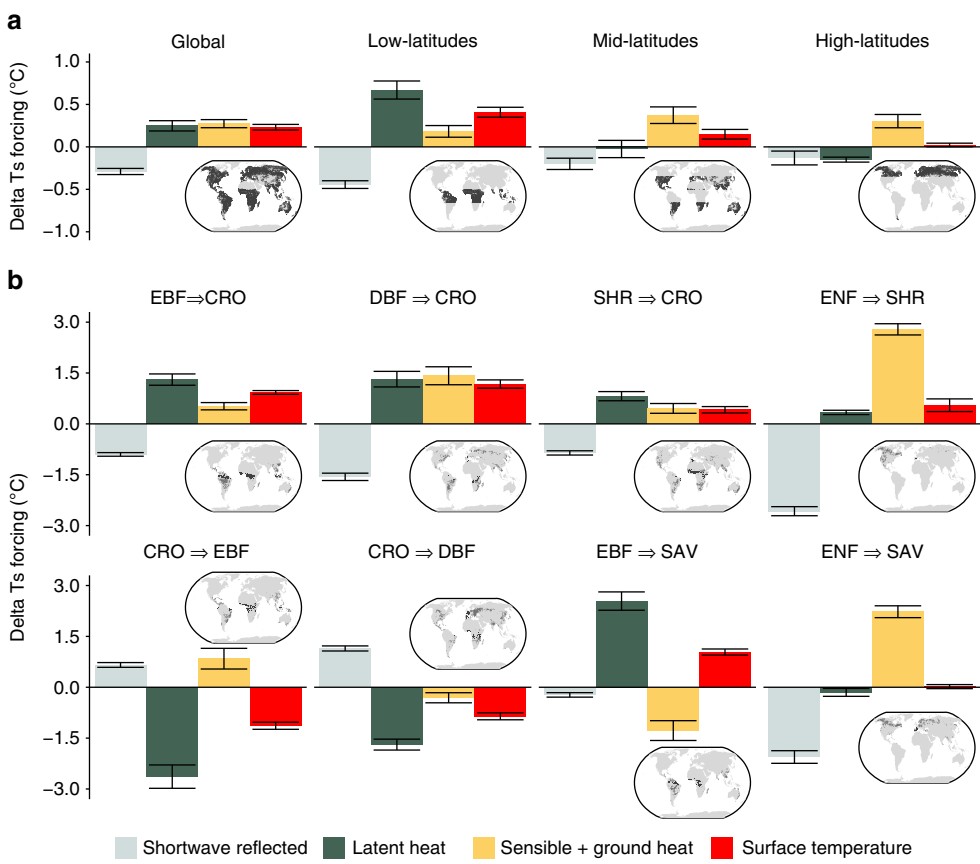

**Fig. 6** Changes in surface temperature resulting from major vegetation transitions between 2000 and 2015. The bars represent the respective contribution of each component of the surface energy balance to the change in surface temperature (Ts). **a** Illustrates changes at global scale or divided by broad latitudinal bands, while **b** shows changes for a selection of transitions types involving the following vegetation classes: evergreen broadleaf forests (EBF), deciduous broadleaf forests (DBF), evergreen needleleaf forests (ENF), savannas (SAV), shrublands (SHR) and croplands (CRO). Each bar represents the mean of all changed pixels weighted by the actual change within each 1° pixel. The errorbars represent ± two times the weighted standard error around the weighted mean

different geographic regions and biomes, confirming the need to jointly assess both radiative and non-radiative processes in order to estimate the changes in surface climate induced by land cover change. In particular, this assessment shows that in ecosystems where vegetation growth is limited by water availability the climate impacts of a vegetation cover transition are dominated by changes in evapotranspiration, whereas in ecosystems where vegetation growth is limited by energy, such as boreal shrublands, the perturbation of the surface temperature is dominated by changes in the radiative and aerodynamic properties of those ecosystems. The origin of actual vegetation cover change in the recent past is divided along the same lines, with direct anthropogenic changes (such as agricultural intensification) occurring mostly where evaporation dominates, while changes within natural ecosystems have generally been confined to higher latitudes where radiative and aerodynamic effects prevail.

Our results show that vegetation cover change over the period 2000–2015 has produced on average a brighter but warmer land surface. This apparently contradictory signal is controlled by the three dominant transitions driven by agricultural expansion in mostly tropical regions (from evergreen broadleaf forests, shrublands or deciduous broadleaf forests to cropland, Fig. 5), which each lead to similar increases in albedo, and consequent reductions in absorbed radiation and turbulent energy fluxes. This perturbation of the surface energy balance ultimately produces a counter-intuitive warming of areas with higher albedo because of stronger plant-mediated constraints on evaporative cooling, in accordance with recent findings that prove the central role of non-radiative biophysical effects mediated by evapotranspiration[18].

In a world that will need to feed more people whilst allocating more land for forestry to serve as a negative emission technology, our results provide the supporting information to assess the overall climate efficiency of alternative land-based mitigation strategies. First, evergreen broadleaf forest is the vegetation type that is most worth preserving in terms of local biophysical effects, as it is associated with the highest potential increases in temperature on transition to all other vegetation types. This provides strong additional support for the United Nations collaborative programme on Reducing Emissions from Deforestation and Forest Degradation in Developing Countries (REDD+), as avoided tropical deforestation is then beneficial for climate mitigation for both biogeochemical and biophysical reasons. Second, conversion to croplands and grasslands consistently leads to local warming irrespective of the original vegetation cover type. This biophysical cost serves as an additional argument against further cropland expansion, as cropland is not only associated with the lowest carbon stocks, but is also a considerable emitter of other greenhouse gases such as nitrous oxide and methane. From both biophysical and biogeochemical points of view, the conversion of evergreen broadleaf forest to cropland appears to be one of the worst land cover transitions for the climate, and yet is the main transition that has occurred in the recent past.

Beyond global level estimates, our results also illustrate that biophysical effects of vegetation cover change vary considerably in geographic and climate space. In fact, the detailed spatial information provided by this study could provide avenues for guiding the development of regional land-based mitigation plans using the natural biophysical properties of the different vegetation types. Regional efforts to combat climate change effects such as desertification, including the Great Green Wall for the Sahara and Sahel initiative, could benefit from the spatially and temporally resolved information that result from our analysis. Ultimately, we expect that our observation-driven methodology could further serve as a baseline to develop monitoring, reporting and verification guidelines for the implementation of land-based climate mitigation and adaptation options for land biophysics, mirroring what is currently done for biogeochemical land processes.

We anticipate that the global assessment presented here, along with the methodology and the freely available data set[30], will generally support the development of land-based plans that target climate mitigation through several applications. First, these observation-based evidences of the role of vegetation on the surface energy balance are an important asset for benchmarking and improving land surface schemes and Earth system models[21]. Being able to tackle directly this local effect is an advantage that observation-driven assessments have over model-based ones, which either have problems disentangling the low land cover change signal from climate noise in their large pixels[19,34], and so have to resort to large-scale idealized simulations in which local and non-local effects are intermingled[12,35], or have to develop extra methodologies to isolate local effects[23,36]. Beyond the evaluation of land surface models, the data set could also help to assess the climate impacts of future scenarios of vegetation cover change within the framework of integrated assessment models, which could harness the spatially and temporally resolved changes in temperature associated with changes in vegetation cover.

The novel methodology developed in this study opens opportunities for further developments. Unlike previous studies[9,11], we describe vegetation using a recent land cover product specifically created for climate studies (ESA CCI land cover maps[32]) and, more importantly, we adopt a methodology that does not require fixing any thresholds in cover fractions to define categorical vegetation classes. Instead, we use the actual cover fraction values for different vegetation types within a grid cell pixel as predictors, thereby separating the different effects of each vegetation type (see Methods section for details). These properties make the method scalable, capable of ingesting layers describing vegetation cover and biophysical properties at any spatial resolution. It could be applied to dedicated studies at regional scales, where more accurate biophysical variables and detailed thematic maps (e.g., describing areas with different land management practices) are available. Despite the increased availability of satellite information at finer spatial resolution, the method can also help to disentangle the signal of vegetation cover change from coarse spatial resolution time series, a prerequisite to exploit the long term data archive of satellite observations available since the 1980s.

It is worth noting that our approach addresses exclusively the direct biophysical impact of land cover change at local scales, since climate feedbacks and large-scale teleconnections cannot be assessed with this method of local space-for-time substitution. However, local effects dominate the overall biophysical impacts when land cover transitions are limited in space[23]. The predominantly local nature of these phenomena has important implications for the implementation of land-based mitigation plans, since it connects the climate impacts to the areas where land policies are implemented. This aspect is of particular relevance for the approval of land-based policies by local communities, that might find in land biophysics additional motivations for protecting and preserving their forests.

## Methods

**Maps of vegetation cover fractions**. The analysis is based on establishing a statistical relationship over a local moving window between vegetation cover fraction maps and variables describing surface properties retrieved from satellite observations (Fig. 7). The vegetation cover fractions are based on the 300 m global land cover map for the year 2010 provided by the European Space Agency's (ESA) Climate Change Initiative (CCI)[32]. The map uses the UNLCCS classification scheme[37], but an open conversion tool is then used both to aggregate and to translate these classes based on user-dependent criteria[38]. This tool was designed to produce maps of the plant functional types typically used in global climate and vegetation modelling. To reach out beyond the modelling community, here we

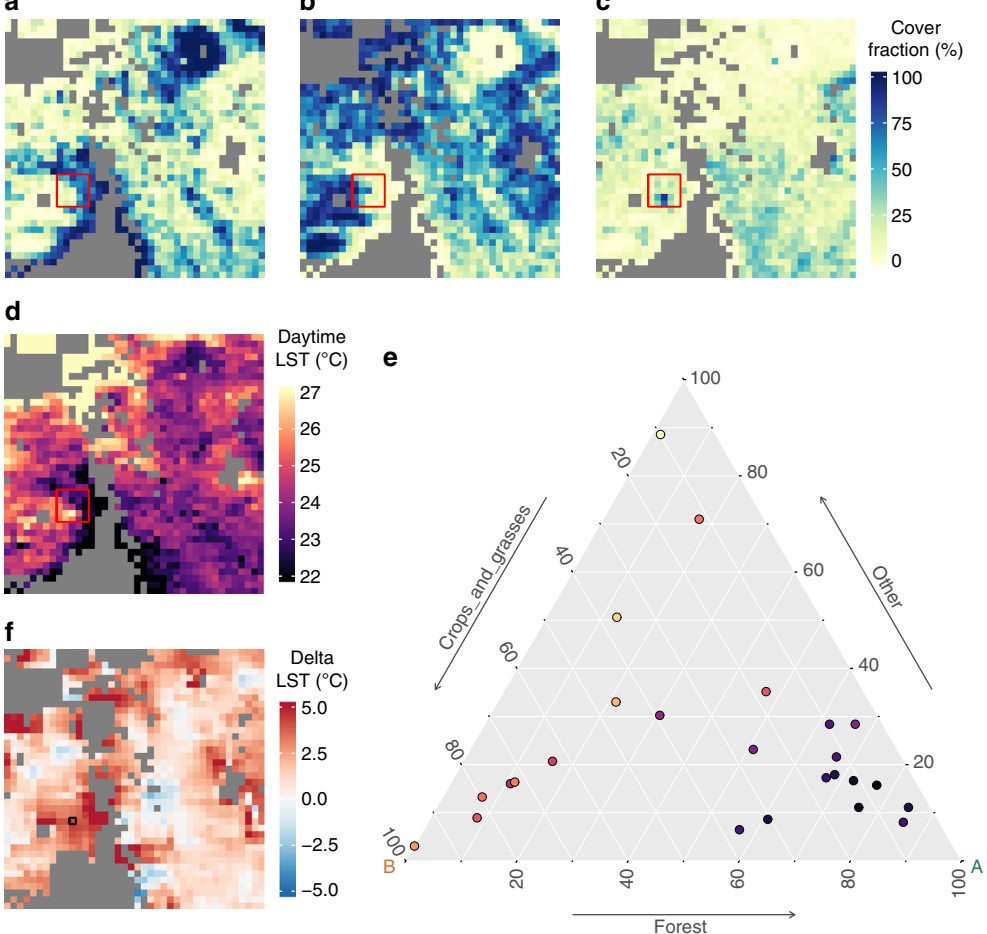

**Fig. 7** Illustration of the methodology for retrieving the local biophysical signal associated to potential vegetation change. The input consists of vegetation cover fraction maps for different classes, in this case: forests (**a**), crops/grasses (**b**) and other (**c**); and the spatial variation of the target biophysical variable, here day-time land surface temperature (LST) (**d**). **e** Ternary diagram showing the values of the 25 pixels in the red boxes of **a**–**c** respectively plotted in the three axes. The colour of the points represents the corresponding values of the biophysical variable. The regression in the compositional space allows the estimation of LST at the vertices of the triangle: vertex A representing a full forest cover and vertex B representing 100% cover of crops/grasses. The difference between these values is the estimated change in LST following a total change from forests to crops/grasses for the pixel in the centre of the 5 by 5 window. **f** The estimated change in LST for the transition from forests to crops/grasses for the entire region. Grey areas are masked out because of the topographical filtering or lack of co-occurrence of either forests or crops/grasses. The size of the pixels is 0.05° or ~5.5 km at the equator

instead used the tool to produce fractions of vegetation classes as defined by the widely used global vegetation classification scheme of the International Geosphere Biosphere Programme (IGBP), based on the conversion matrix presented in Supplementary Table 1.

**Data preparation.** The input surface property variables need to have a common spatial resolution (0.05°), temporal coverage and monthly temporal resolution. These variables are all derived from post-processing of remote sensing products based on measurements from the NASA Moderate Resolution Imaging Spectro-radiometer (MODIS) instrument on-board of the Aqua and Terra satellites. For all variables, the median value for each month is calculated from the years 2008 to 2012 to generate the 5-year climatology while retaining the seasonal cycle. The source and pre-processing of each individual variable is as follows.

Day-time and night-time LST are the radiant temperature of a surface measured in the day or at night, respectively. The MODIS instrument on board the Aqua platform makes such measurements twice over its cycle at ~13:30 and 1:30 local time at the Equator. These times are close to those at which the minimum and maximum temperatures are expected. The MODIS land surface temperature algorithm provides such estimates at a monthly time step at 0.05° spatial resolution[39,40] under the product MYD11C3 (we use collection 5 available from the NASA LPDAAC website https://lpdaac.usgs.gov/).

Surface upwelling longwave radiation is the outgoing infrared radiation emitted by the surface. A large part of this energy is absorbed by the atmosphere and later re-emitted towards the Earth (by clouds and greenhouse gases) or outwards to space. The upward longwave radiation (LW↑) can be calculated from the surface

temperature ($T$) and broadband emissivity ($\epsilon_B$) using the Stefan-Boltzmann law:

$$LW_{\uparrow} = \epsilon_B \sigma T^4 \qquad (1)$$

where $\sigma$ is the Stefan-Boltzmann's constant ($5.67 \times 10^{-8}$ W m$^{-2}$ K$^{-4}$). We use day-time and night-time LST from the MYD11C3 product to estimate the mean surface temperature ($T$) over the entire day span using a simple average. The MYD11C3 product also provides emissivity estimates for various specific narrow bands in the middle and thermal infrared spectrum that can be used to obtain $\epsilon_B$ using the empirical equation suggested by a dedicated study[41]:

$$\epsilon_B = 0.2122\epsilon_{29} + 0.3859\epsilon_{31} + 0.4029\epsilon_{32} \qquad (2)$$

where $\epsilon_{29}$, $\epsilon_{31}$ and $\epsilon_{32}$ are the estimated emissivities in MODIS bands 29 (8400–8700 nm), 31 (10,780–11,280 nm) and 32 (11,770–12,270 nm). Because the satellite can only measure during cloud-free observations, it must be specified that the resulting monthly upwelling longwave radiation only refers to clear sky conditions, which we will denote using an asterisk as LW$_{\uparrow}^*$.

Albedo is defined as the proportion of the incident light or radiation that is reflected by a surface. In this case, we are interested in the monthly average of the proportion of total radiation across the broadband shortwave spectrum reflected by the Earth's surface every 0.05°. The NASA MCD43C3 albedo product provides 8-daily estimates of both directional hemispherical albedo (black-sky albedo) and bihemispherical albedo (white-sky albedo) based on multidate multispectral MODIS cloud-free observations collected over a 16-day moving window and a semi-empirical kernel-driven bidirectional reflectance model[42]. These white-sky and black-sky albedos correspond to theoretical situations in which incident

radiation is either completely diffuse or completely direct. To obtain an estimate of real conditions without information on the fraction of diffuse radiation, we took the mean of both values. To have estimates at monthly temporal resolution, we selected only those in which the 16-day periods correspond best with the 15th of each month. The data are available from the NASA LPDAAC website (https://lpdaac.usgs.gov/).

Latent heat flux is the flux of heat from the Earth's surface to the atmosphere that is associated with evaporation of water at the surface. In this case, we are interested in the terrestrial component associated with plant transpiration. The MOD16A2 product[43] provides latent heat obtained by integrating several MODIS products (land cover, albedo, leaf area index, and fAPAR) with meteorological data, delivered at 0.05° spatial resolution with monthly temporal resolution covering the regions from 60°S until 80°N. The product is not entirely observation-driven as it requires some specific parametrization per biome, but which is not spatially explicit. The data are available from the NTSG website (http://www.ntsg.umt.edu/project/mod16).

**Retrieving the local biophysical signal of vegetation change**. To identify the biophysical signal due to changes in vegetation cover we establish a relationship between vegetation cover fractions and the surface variables over a local moving window. As a result of this, the direct biophysical effects of vegetation change considered here are local. This is valid both for the spatial extent of the cover change, which assumes at most a change of a complete fine resolution pixel (0.05° × 0.05°), and for the origin of the change, i.e., we ignore indirect effects due to regional change from neighbouring areas. The moving window size is 5 by 5 pixels at 0.05° resolution, covering an area of ~25 km by 25 km over which the local climate is assumed to be uniform. To unmix the signal generated from the compositional land cover, for each window we apply a linear regression using a matrix $\mathbf{X}$ containing the vegetation fractions of each of the 25 pixels as explanatory variables and a vector $y$ containing the 25 values of a given biophysical variable as response variable to obtain a vector of $\beta$ coefficients:

$$\mathbf{y} = \mathbf{X}\beta \qquad (3)$$

This is equivalent to solving the following system of equations:

$$\begin{cases} y_1 = \beta_1 x_{11} + \beta_2 x_{12} + \ldots + \beta_m x_{1m} \\ y_2 = \beta_1 x_{21} + \beta_2 x_{22} + \ldots + \beta_m x_{2m} \\ \vdots \\ y_n = \beta_1 x_{n1} + \beta_2 x_{n2} + \ldots + \beta_m x_{nm} \end{cases} \qquad (4)$$

in which $x_{ij}$ represents the cover fraction of vegetation $j$ in pixel $i$, for the $n$ pixels in the moving window and the $m$ classes that are considered. Once identified, we can use the $\beta$ coefficients to predict the local $y$ value corresponding to a given composition $x$, including that composed of a single vegetation cover $j$ by setting $x_j = 1$ and all other $x$ values to zero.

There is a problem, however, if the compositional predictor data set $\mathbf{X}$ is used directly in the analysis. Compositional data can behave somewhat differently to 'ordinary', open or normal data, because compositions necessarily sum to one (for this reason they are also sometimes described as 'closed' data). Statistically, this can lead to spurious correlations between compositional components, and/or between compositional components and the response variable. Analysis of any given subset of compositional components can lead to very different patterns, results and conclusions[44]. Geometrically, all points defined by the compositions must fall in a simplex because their compositions sum to one. For a three part composition, this simplex is a triangular plane (i.e., it exists on a 2-dimensional surface, such as panel **e** in Fig. 7). While the matrix has 3 columns, there are only (at most) 2 dimensions. A transformation of $\mathbf{X}$ is needed to reduce appropriately the dimensionality of this matrix for subsequent use in the regression.

The transformation we apply to reduce the dimensionality of $\mathbf{X}$ involves a singular value decomposition (SVD). This procedure is very close to a principal component analysis (PCA). The first step consists of centring all the columns of the predictor matrix $\mathbf{X}$ of vegetation fractions by removing the column means. We then apply the SVD:

$$(\mathbf{X} - \mathbf{M}) = \mathbf{U}\mathbf{D}\mathbf{V}^t \qquad (5)$$

where $\mathbf{M}$ is the appropriate matrix of column means, $\mathbf{U}$ and $\mathbf{V}$ are the matrices containing respectively the left hand and right hand singular vectors, and $\mathbf{D}$ is a diagonal matrix containing the singular values (the standard deviations of the ensuing dimensions). Squared values of $\mathbf{D}$ indicate how much variance is explained by each (orthogonal) dimension. We implement a rule where as many dimensions from this SVD are retained as to conserve 100% of the original matrix's variation. In doing so, we reduce the dimensionality appropriately as described above, as well as remove what may be additionally redundant dimensions that can occur locally if, for instance, the only points in which 2 classes are represented have exactly the same values. To avoid having problems when there is too little or no information (e.g., if all pixels have exactly the same compositions), we added a pre-condition

that there must be at least 10 pixels with different compositions. The final appropriately transformed predictor matrix of reduced dimension $\mathbf{Z}$ is then obtained by:

$$\mathbf{Z} = (\mathbf{X} - \mathbf{M})\mathbf{V}_z \qquad (6)$$

where the subscript $z$ in $\mathbf{V}_z$ indicates that the latter is composed of a subset of right hand singular vectors in $\mathbf{V}$ as selected from $\mathbf{D}$ as described above. The resulting predictor matrix $\mathbf{Z}$ can now be regressed onto the local biophysical variable $y$.

$$y = \mathbf{Z}\beta + \varepsilon \qquad (7)$$

where $\mathbf{Z}$ has been augmented with a leading column of ones to accommodate an intercept term in the regression. The standard manner to obtain an estimate of $\beta$ is:

$$\beta = (\mathbf{Z}^t\mathbf{Z})^{-1}\mathbf{Z}^t y \qquad (8)$$

Because the compositional predictor matrix $\mathbf{X}$ has been transformed to matrix $\mathbf{Z}$, regression coefficients identified in the regression of $\mathbf{Z}$ onto $y$ do not immediately provide information about the association between the various vegetation cover fractions and the surface property variables. In order to identify the $z$ values associated with a particular vegetation (in that local analysis) we instead define a 'dummy pixel' whose composition contains only that vegetation class, with all other classes in the dummy pixel's composition set to zero. This pixel's composition is then transformed, and its $y$ value predicted. This is the $y$ associated with that vegetation type. Since we wish to do this for all compositional components of interest, we actually define a matrix $\mathbf{P}$ with as many rows as these compositional components that we wish to predict. $\mathbf{P}$ is centred on the same column means as above ($\mathbf{M}$, specific to each local analysis), and then multiplied by the correct number of transposed right hand singular vectors ($\mathbf{V}_z$, again, specific to each local analysis).

$$\mathbf{Z}_p = (\mathbf{P} - \mathbf{M})\mathbf{V}_z \qquad (9)$$

Predicted $y_p$ values for each vegetation type (identified by predicting the appropriately transformed 'dummy pixels') are then calculated as:

$$y_p = \mathbf{Z}_p\beta \qquad (10)$$

The expected change in variable $y$ associated with a transition from one vegetation type to another at the central pixel of the local window is then the difference between the $y_p$ predicted for each pure vegetation type:

$$\Delta y_{A \to B} = y_B - y_A \qquad (11)$$

Beyond our primary interest in the change $\Delta y$ for a given vegetation transition, we also assess the uncertainty associated with each of these differences. We consider uncertainty in terms of standard deviations, and thus, according to error propagation, the uncertainty for the difference due to the transition from A to B can be determined from:

$$\sigma_{A \to B} = \sqrt{\sigma_A^2 + \sigma_B^2 - 2\sigma_{AB}} \qquad (12)$$

where $\sigma_A^2$ and $\sigma_B^2$ are the variances in the estimates of $y$ for each vegetation type, and $\sigma_{AB}$ is their covariance. This covariance term is important as the uncertainties of the individually predicted $z$ values are not independent given that they derive from the same regression model. The variances and covariances of all vegetation types can be obtained from the covariance matrix, which in turn is calculated as:

$$\Sigma = \mathbf{Z}_p \text{Var}[\beta]\mathbf{Z}_p^t \qquad (13)$$

The diagonal terms in $\sum$ are the variances of individual predictions of (individual) vegetation classes. The off-diagonal parts of $\sum$ hold the covariances between these predictions.

The whole procedure described above (variable transformation, regression and uncertainty estimation) is applied globally over 5 by 5 moving windows for the 3 biophysical variables for each of the 12 months of the year at 0.05° spatial resolution for each vegetation transition considered. Symmetric transitions yield identical results (e.g., $\Delta y_{A \to B} = -\Delta y_{B \to A}$). The resulting maps only provide information for the pixels in which all 25 pixels in the moving window had information.

**Masking out sub-optimal conditions**. The method relies on there existing co-occurrences of vegetation classes within the local window. Furthermore, the statistical methods that are applied to these sets of points are more likely to provide reliable results when there are large and balanced presences of both vegetation classes of interest. An index quantifying mutual presence ($I_c$) is thus applied for each pair of vegetation classes (see our accompanying data paper for more details[30]), and a threshold of $I_c < 0.5$ is used to mask out from the results those pixels whose local windows do not provide enough co-occurrences.

Another masking operation is required to remove areas where high topographical variability exists within the local window. Topographical relief generally translates into climatic gradients, which would compromise the space-for-time approach. Pixels are masked according to three criteria: (1) standard deviation of elevation within the local window must remain below 50 m; (2) the difference between the mean elevation of the central pixel and the mean elevation of the entire local window should be less than 100 m; and (3) the difference in the standard deviation of elevation within the central pixel and that over the entire local window should remain below 100 m. For more information on this masking step, readers are again directed to the data descriptor paper[30].

**Spatial aggregation**. The maps resulting from the local space for time analysis need to be spatially aggregated from 0.05° to 1° grid cells to be used alongside data from the CERES instrument, which provides the information necessary to close the energy balance. Aggregating to 1° also has other advantages, namely: (1) a mean difference of a variable associated with change from one vegetation type to another may be assumed to be more accurate than any individual estimate at finer scale; (2) this scale is simpler to map and visualize at global level; and (3) it is more comparable to results from land surface models. Because each 0.05° estimate of $\Delta y$ includes an associated estimate of its uncertainty, this uncertainty can be used to down-weight less reliable values during the aggregation procedure. The typical approach to do so is weighting based on the inverse of the uncertainty:

$$\overline{\Delta y} = \frac{\sum_i \Delta y_i / \sigma_i^2}{\sum_i 1 / \sigma_i^2} \tag{14}$$

where $\overline{\Delta y}$ is the mean aggregated value, whose uncertainty is calculated as:

$$\sigma_{\overline{\Delta y}}^2 = \frac{1}{\sum_i 1 / \sigma_i^2} \tag{15}$$

However, these formulations do not account for the spatial auto-correlation generated by the moving window (1 to 20 pixels may be common between two nearby estimates depending on the possible overlap of their respective 5 by 5 windows). This auto-correlation problem may be compounded further when only a clustered set of 0.05° samples are available within the 1° by 1° area. This can occur due to the topographical masking, or because two vegetation types only co-occur over a small part of the 1° grid cell.

To tackle this auto-correlation, we employ a more generic weighting approach. The weights depend not only on the uncertainties estimated from the regressions as above, but also on how each window is correlated with every other window within the area of 1°. This information is summarized in a 400 by 400 matrix $\mathbf{R}_a$ containing the fraction of overlap between every pair of windows. The information in $\mathbf{R}_a$ is combined with that of the pixel-wise uncertainties that are embedded in $\mathbf{D}_a$, a diagonal matrix containing the uncertainties in its diagonal, to build a covariance matrix $\sum_a$ (the subscript $a$ is used to differentiate these matrices involved in this aggregation step from those used before):

$$\sum_a = \mathbf{D}_a \mathbf{R}_a \mathbf{D}_a^t \tag{16}$$

The vector of weights is then obtained as:

$$\mathbf{w} = \frac{1}{\mathbf{1}^t \Sigma_a^{-1} \mathbf{1}} \Sigma_a^{-1} \mathbf{1} \tag{17}$$

which can then be used to calculate the aggregated $\overline{\Delta y}$ as:

$$\overline{\Delta y} = \sum_i w_i \Delta y_i \tag{18}$$

while the aggregated uncertainty $\sigma_{\overline{\Delta y}}^2$ is given by:

$$\sigma_{\overline{\Delta y}}^2 = \mathbf{w}^t \Sigma_a \mathbf{w} = \frac{1}{\mathbf{1}^t \Sigma_a^{-1} \mathbf{1}} \tag{19}$$

When the windows have no auto-correlations, both Eqs. (18) and (19) fall back to the simpler weighting formulas of Eqs. (14) and (15). The aggregation procedure is applied to all data layers.

**Detection and treatment of outliers**. Despite all efforts to characterize uncertainty and reach representative values, the results can still contain unrealistic values. A reason for this might be that uncertainties in the input data (the remote sensing biophysical variables and the vegetation cover fraction maps) are not explicitly taken into account. As a final step to remove possible outliers, we remove all values for grid cells in which there are not at least 20 samples at 0.05° spatial resolution. Lastly, we also remove values that are statistical outliers based on the distribution of the entire data set. All data layers are available with their associated uncertainty. Supplementary Fig. 5 illustrates the pixels for each vegetation transition where data is available.

**Closing the surface energy balance**. The local unmixing step can only be applied to those variables available at the 0.05° spatial resolution (namely $\alpha$, LE, LW$^*_\uparrow$, LST$_{day}$ and LST$_{night}$), meaning some components of the surface energy balance are missing. The full surface energy balance is expressed as:

$$SW_\downarrow - SW_\uparrow + LW_\downarrow - LW_\uparrow = H + LE + G \tag{20}$$

SW$_\downarrow$, SW$_\uparrow$, LW$_\downarrow$ and LW$_\uparrow$ are respectively the downwelling and upwelling radiative fluxes in the shortwave or longwave parts of the spectrum, LE is the latent heat flux, $H$ is the sensible heat flux and $G$ is the ground heat flux. We derive the terms of the energy balance combining MODIS-based data sets with the EBAF-Surface Product derived from the NASA Clouds and the Earth's Radiant Energy System (CERES) instrument. This data set (CERES EBAF-Surface Ed2.8) provides a closed and gap-filled surface energy balance at 1° spatial resolution that is consistent with CERES top-of-atmosphere irradiance measurements[45]. For the specific goals of this analysis we are interested in how the terms of this equation change according to a change in vegetation cover, i.e.,:

$$\Delta SW_\downarrow - \Delta SW_\uparrow + \Delta LW_\downarrow - \Delta LW_\uparrow = \Delta H + \Delta LE + \Delta G \tag{21}$$

We make the assumption that the changes in vegetation cover that are considered here are too small (i.e., maximum 0.05°) to generate strong feedbacks in the cloud regime, and as a consequence we assume $\Delta SW_\downarrow = 0$ and $\Delta LW_\downarrow = 0$. The change in reflected shortwave radiation can be expressed in terms of albedo ($\alpha$) and incoming shortwave radiation ($\Delta SW_\uparrow = \Delta \alpha \times SW_\downarrow$), the latter being available from CERES data at 1° resolution. Although we derived estimates of changes in upwelling longwave flux satellite measurements at 0.05°, these refer to clear sky conditions only (i.e., when the satellite instrument can measure the ground unobstructed by clouds) while other fluxes are representative of all cloud conditions. As a proxy for the effect of cloudiness, we used a correction factor based on the ratio of all sky (LW$_{C\uparrow}$) to clear sky $\left(LW^*_{C\uparrow}\right)$ longwave upwelling estimated by CERES ($\Delta LW_\uparrow = \left(\Delta LW_{C\uparrow}/LW^*_{C\uparrow}\right) \times \Delta LW^*_\uparrow$, where the asterisk indicates values for clear sky conditions). By re-writing and simplifying the equation above, the expression describing the change in the residual flux, composed of both sensible and ground heat fluxes, becomes:

$$\Delta(H + G) = -(\Delta \alpha) SW_\downarrow - \left(\Delta LW_{C\uparrow}/LW^*_{C\uparrow}\right) \times \Delta LW^*_\uparrow - \Delta LE \tag{22}$$

We apply this expression to every 1° pixel for every month of the time series and every vegetation transition based on the previously calculated data sets of $\Delta \alpha$, $\Delta LW^*_\uparrow$ and $\Delta LE$. To have all terms of the energy balance on equal footing and with the same sign convention, we also explicitly produced data sets of shortwave reflected radiation ($\Delta SW_\uparrow$) and full-sky longwave emitted radiation ($\Delta LW_\uparrow$), all of which are freely available in the data repository[30].

**Controlling for consistency in latent heat flux**. Although a validation of the methodology against ground-based measurements of surface energy balance fluxes would be desirable, no adequate network of measurements currently exists. Flux-towers would constitute the right measurements, but we would need a large and well distributed number of paired flux-tower sites with contrasting vegetation types yet similar climate, which currently do not exist. A comparison over a handful of sites[30] indicate that the results are in the right direction, even if the low number of pair-sites does not allow a robust and comprehensive verification of our data set.

In the absence of validation, we propose a diagnostic to evaluate the robustness of latent heat flux (MOD16A2), arguably the most questionable input product, against an alternate data-driven product GLEAM v3.1[46,47]. We cannot directly use GLEAM in a space-for-time approach because its spatial resolution (0.25°) is too coarse to ensure homogeneous climate conditions within the moving window. However, by aggregating both MOD16A2 and GLEAM to a common 1° spatial resolution and comparing the latent heat flux for all pixels with high forest cover (>75%) against those with low forest cover (<25%), we come to the conclusion that the capacity of MODIS MOD16A2 product to discriminate differences due to forest cover are adequate for our methodology and scope (Supplementary Fig. 6).

**Calculating the effects of past vegetation changes**. The changes in land cover from 2000 to 2015 are obtained from their respective maps from the ESA CCI products. Both 300 m land cover maps are converted to vegetation fraction maps using the same methodology[38] and table (Supplementary Table 1) as before, but setting the output spatial resolution to 1° instead of 0.05°. We then subtracted the resulting vegetation fractions of 2000 from those of 2015, yielding net changes at 1° per vegetation type. To obtain changes from one vegetation type to another, for each 1° grid cell we match the net changes in vegetation cover experiencing a loss to those other vegetation covers experiencing gains. We assume here that gains originate equally from all vegetation types suffering a loss. With the gridded values of actual vegetation change, it is then possible to multiply them with the corresponding potential changes, $\Delta y$, in order to obtain actual changes for each flux in

the energy balance. The assumption here is that the biophysical effects of vegetation cover change estimated from the 2008–2012 period remain valid for the 2000–2015 period. The resulting values are integrated in time to provide an estimate in exaJoules ($10^{18}$ Joules) of the energy change caused by the given vegetation cover transitions.

**Decomposition of the surface energy balance.** The surface energy balance can be decomposed to isolate the respective contribution of each component to the surface temperature resulting from the vegetation cover transition[24]. Such a method has been used to separate direct contributions (e.g., surface reflection and evapotranspiration) from indirect contributions due to atmospheric feedbacks (e.g., cloud-radiative feedbacks) from global climate simulations[48,49]. While only the direct effects can be separated in our data, it provides an easier way to interpret the energy effects caused by vegetation cover change from 2000 to 2015. To do so, we use Eqs. (1) and (20) and rearrange the terms as:

$$\varepsilon_B \sigma T^4 = SW_\downarrow + LW_\downarrow - SW_\uparrow - LE - (H + G) \quad (23)$$

Calculating the derivative of Eq. (23) to represent the change in vegetation cover, neglecting changes in emissivity and isolating for temperature, we obtain:

$$\Delta T = \frac{1}{4\sigma T^3}\left(SW_\downarrow + \Delta LW_\downarrow - \Delta SW_\uparrow - \Delta LE - \Delta(H + G)\right) \quad (24)$$

The changes in downwelling radiation are neglected as before, leaving the changes in surface reflection, in latent heat and in the residual fluxes. To connect with the global actual energy changes from 2000 to 2015 calculated before, the latter are transformed back into fluxes that can be used in Eq. (24). To do so, the energy values in Joules are divided by the total changed area for each transition, and divided again by the number of seconds in a year, to obtain estimates of the annual radiative forcing at the surface that these changes have caused.

**Assumptions and limitations of the study.** A number of assumptions were necessary to make our assessment which need to be taken into account when interpreting the results. To close the surface energy balance locally, we assume that the local cover change at 0.05° does not generate systematic changes in cloud cover between the grid cell of the moving window of 0.25° and therefore affect indirectly the local surface energy balance. The robustness of this assumption relies on the fine scale of the analysis and on the typical lateral movement of air masses due to wind that ultimately advect air masses and clouds to different grid cells. Note that the final upscaling to 1° spatial resolution still represents an average effect of land cover change at 0.05° resolution, but now smoothed over that 1°. We also assume vegetation cover is the only driver of changes in surface biophysics within the local moving window of 0.25°. For this purpose areas with strong elevation gradients are masked out to filter topographic effects, whereas any spatial gradient in general soil properties within the moving window is not considered. In our analysis of past changes, we consider that the biophysical signal of land cover change derived from observations acquired in 2008–2012 are representative for the entire 2000–2015 period. This assumption holds if the background climate does not change substantially[50], and could further be used to explore biophysical impacts in the near-future, but would require special attention to projecting them in a changing climate (e.g., places with current snow cover in spring would lose the strong seasonal albedo feedback if temperatures rise substantially). This assumption relies on the dominant role that climate variability has on climate trends on the decadal time scale. Finally, the biophysical effects considered here result from analyzing average conditions over 5 years (2008–2012), while changes in more extreme years might be amplified.

**Data availability.** The numbers presented in this assessment are averaged both temporally and spatially, hiding part of the wealth of information generated in the underlying data. The original data set reports for each vegetation transition, all changes in the surface energy balance at a monthly scale over a 1° × 1° spatial grid for all places where two vegetation classes co-exist. Every record in space and time is also accompanied by an estimate of the uncertainty associated with the methodology that can serve to assess the relative quality of each value. To fully appreciate the depth of the information generated, readers are redirected to the accompanying data descriptor[30], and to the actual data set, which is freely available in the Figshare repository, https://doi.org/10.6084/m9.figshare.c.3829333.

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

## Acknowledgements

The study was funded by the FP7 LUC4C project (grant no. 603542). We thank NASA for providing the MODIS LST and albedo data (via the LPDAAC); the NTSG group of the University of Montana for providing the MODIS ET data; ESA and the Université catholique de Louvain for making available the ESA CCI land cover data and user tool; and the University of Ghent for making available the GLEAM data set. All calculations and plots have been realized with R, using the ggplot2, dplyr, raster and ggtern packages. Maps are made with Natural Earth. Special thanks to D. Fasbender who gave invaluable advice on mathematical aspects of the method.

## Author contributions

G.D., J.H. and A.C. conceived and designed the study, G.D. analyzed the data, G.D. and A.C. interpreted the results and wrote the manuscript with contributions from J.H.

## Additional information

**Competing interests:** The authors declare no competing financial interests.

