## [Peer Review File · Nature Communications]

Editorial Note: This manuscript has been previously reviewed at another journal that is not operating a transparent peer review scheme. This document only contains reviewer comments and rebuttal letters for versions considered at Nature Communications. Mentions of prior referee reports have been redacted.

Reviewers' comments:

Reviewer #1 (Remarks to the Author):

Duvailler et al. presents a very good synthesis of the biophysical energy response to recent land use change based solely on satellite observational changes. The analysis is solid, the paper is well written, and the focus on global energy budgets helps drive this research topic forward.

The only comment I have is that it would be useful to highlight the total energy changes by land use type (e.g. J/year globally for each of the land use classes). This could be done in a table and would further highlight the oversized role of deforestation to agricultural conversion on biophysical climatic feedbacks. If space is limiting, maybe put Figure 3 in the supplemental

Reviewer #2 (Remarks to the Author):

This study presents a typical satellite-based exercise to characterise the potential or actual biophysical controls imposed by vegetation across the globe. Seems like that this work is a continuation and expansion on the Science paper [Biophysical climate impacts of recent changes in 365 global forest cover] by one of the co-authors. Overall, I think the authors did a pretty good job.

Technically speaking, the data processing and analyses appear reasonable. Given the nature of satellite measurements and many potential confounding effects and sources of uncertainty, I think it is hard to justify one approach is better than another, so here I refrain from commenting on or criticizing the technical specifics. As long as the authors acknowledge these limitations and offer caveats, I, as a reader, am happy and comfortable with the findings

From a story/implication side, I feel the authors can spend more efforts figuring out some key messages you want readers to take away; otherwise, my current reading is that they just create some information valuable to assess land cover changes--well, I appreciate this point but am left unsatisfied.

No other specific comments except a minor comment on Fig.4. There, it will do readers a service if explaining what the small circles mean. Do it mean no changes or insignificant testing?

Reviewer #3 (Remarks to the Author):

Manuscript Review: "The Mark of Vegetation Change of Earth's Surface Energy Balance"

This manuscript presents an analysis of changes in the surface energy balance as a result of different transitions in land cover. The analysis is based upon a combination of existing satellite datasets and derived products by the authors. The emphasis of the paper include: a decrease in ET when considering all vegetation change and that they use an innovative method to derive a dataset of the full surface energy balance to examine vegetation transition dependent changes in the surface energy balance. While the concept of examining the dependence of the surface energy

balance to different land cover transition types is novel and a worthy contribution I feel that by focusing the main conclusions on all vegetation change the authors detract from the land use transition type dependence of their results. Furthermore, it is not clear what the main take home message is from the abstract. The manuscript reports the creation of a new global dataset of the full surface energy balance that could be used for validation of Earth System Models. There are a couple of issues with this that I think are critical for publication. See major comments for details. Generally the manuscript is well written and with further evaluation to check the robustness of the methodology of the derived surface energy balance components I do think the research would make a valuable contribution to the field if the emphasis remains on the vegetation transition types and not the aggregated values.

Major Comments

- While datasets that include the full surface energy balance (SEB) at a global scale are scarce I feel that there are some issues with how this dataset has been created.
- SWup and H are derived from the available albedo, ET and LWup and CERES datasets. Even though the albedo, ET and LWup and CERES datasets have been validated as clear in their respective publications, the two new components (SWup and H) should go a thorough validation to against other observational datasets (e.g. FLUXNET-MTE for H – see Jung et al. 2011 doi:10.1029/2010JG001566) to evaluate if the derivation method is robust to the assumptions that are made about G, SWdown and LWdown. In particular, if the idea is to propose that ESM modelers can use this dataset for validating their models, they will assume that the dataset creators have done this validation. Therefore it is critical that the authors show that their approach to derive the full SEB is valid, particularly if the intention is to share this dataset with model developers.
- I was not convinced that the assumption to disregard changes in SWdown, LWdown and G is valid. It assumes that one can ignore the effects of cloud cover over a 1° resolution despite a cloud cover 'correction' applied later to LWup. For the resolution to which SWup and H are derived, assuming zero changes in G...I don't think this can hold at that resolution. I think the authors need to provide a more compelling argument on why these assumptions are reasonable. I also don't understand why the authors don't use SWdown and LWdown from the CERES EBAF-Surface dataset...these variables are included and if SWup and LWup are used so the assumption that these two terms can be ignored doesn't quite make sense if the CERES dataset has them.
- I got the impression that the dataset covers a short time (2008-2012); I'm not sure how useful this is for ESM validation. So perhaps better to avoid overselling that this is the dataset all modelers should use for validation or provide a more compelling case on why this dataset is better than LandFlux, FLUXNET-MTE, MERRA or ERA-Interim.
- Analysis of the LST data between 2000 and 2010. How certain are the authors that the changes in temperature can be directly attributed to land use transitions and not climate change? I know that perhaps one can discount this due to examining differences in the SEB between vegetation types in the 5x5 grid box at the same time for the 2010 (2008-2012 epoch analysis) but I think this is missing from the manuscript.
- The current structure of the manuscript could be refined by starting with the global all vegetation transition types together, global maps of the forest to grass transition and then the changes in SEB split according to the 7 transition classes. This would demonstrate how much more information is gained by refining the examination of changes in SEB as opposed to aggregating the results to all vegetation change. I think this would make the manuscript more logical.
- What could be really nice is to invoke a surface energy balance decomposition with the new dataset (see examples in Luyseart et al. 2014, doi:10.1038/nclimate2196; Thiery et al. 2017, doi:10.1002/2016JD025740; or Hirsch et al. 2017, doi: 10.1002/2016JD026125) to understand which component of the surface energy balance explains the reason for temperature change for the different transition types. This could really improve the impact of the manuscript.
- I find that the methodology description in both the main manuscript and the supplementary is thorough. There are occasions where perhaps more information or a statement justifying a particular decision /assumption would be desirable which I have noted where I feel this is

necessary in the minor comments below.

- If possible, discussing the results further in the context of previous literature involving SEB dependence on the vegetation transition type would be desirable.

Minor Comments

- The title doesn't peak my interest perhaps changing to something along the lines of "Perturbation of the Surface Energy Balance Depends on Vegetation Transition Type" would be more apt.
- Lines 22-25: This statement refers to all vegetation change, but Line 14 alludes to the need to consider the type of vegetation change. Given what is also shown in Figure 2, it would be nice if this could be more specific to a type of vegetation change (e.g. trees to grass) rather than all vegetation change (which can include trees to grass but also grass to trees). Perhaps it would also make a stronger statement to include how much of the land area has changed over this period
- Line 25: "The freely available dataset will provide valuable information to assess future scenarios of land cover change". I'm not sure this is the best place to put such a statement because the existing narrative doesn't really discuss how this could be done apart from as a tool to evaluate whether ESMs get the 'right' changes in the SEB with the different vegetation transition types...but then how can we be certain what the transitions in the future will be?
- Line 38: Replace "in the reflectance of solar radiation" with "albedo"
- Line 46: Reference 7 is also relevant here
- Line 48: Replace "partition" with "partitioning"
- Line 58: Replace "land's" with "the terrestrial"
- Line 59: Delete "one of"
- Line 60: Replace "With the first the signal is identified" with "The first identifies the signal"
- Line 63: Replace "alternative" with "second"
- Line 69: Here the authors introduce the concept of "multiple vegetation transitions" and I think this could be emphasized more than it is in the abstract because this is where the research really is novel.
- Line 74: "Our vegetation is defined by continuous fractions instead of crisp classes" what does this mean. Please rephrase to clarify.
- Lines 76-79: The authors refer to using 'mixed pixels' and then the 'un-mixing methodology'...please clarify
- Line 81: surface energy balance
- Line 82: "for the epoch 2008-2012" but later on the authors compare 2000 and 2010 changes. This is a bit confusing. Furthermore if the intention is to make this data available for ESM the validation the period is short.
- Line 88: "the three class scheme" could be removed here as this is explained in the supplementary material or include a statement on what this means here.
- Line 89: But also over South East Asia which generally isn't considered a water limited area but an energy limited area (see Seneviratne et al. 2010; doi:10.1016/j.earscirev.2010.02.004)
- Line 86-104: The narrative here starts with Figure 1e and f and then moves to Figure 1a-d. Perhaps it would be more logical to rearrange the panels of Figure 1 to match the narrative
- Line 91: The references 13,14 here seem unnecessary
- Line 95: Include reference for "as grasses are typically brighter than trees" – even if its common knowledge
- Line 99: "being dominated by the reduction in sensible heat in cold and/or humid climates at northern latitudes" – be careful here because over South East Asia and India there is an increase in H and large decrease in LE.
- Line 105: As earlier on line 88 the authors don't define "the seven class scheme" please refer to where this is defined in the supplementary and avoid confusing the reader.
- Line 110: "decade 2000-2010" contrasting period to that reported on line 82. It's easy for the reader to get confused if two methods are used that cover different periods 2008-2012 for the 'space-for-time' logic and 2000-2010 for the other. Switching between the two here is confusing.
- Line 114: It looks like the effects of TrBrEv to ManGra is larger than TrBrEv to NatGra.
- Line 117: Unless I've read this wrong from Figure 2 but TrBrDe to NatGra results in an almost 10

W/m² change in LE and is larger than H. Are the authors perhaps referring to TrNeEv to NatGra here?

- Line 118: “whilst the seasonality of LE after the transition to cropland markedly follows that of the boreal summer” – I don’t think this adds any value here unless the authors have a particularly point they would like to make that this transition type is mostly occurring in the Northern Hemisphere.
 - Line 127: “that proved the fundamental importance of biophysical processes on the long-term efficacy of land-based mitigation in Europe” – Doesn’t seem fitting to have this here, the authors haven’t shown this in their results and the focus of the manuscript is not on mitigation.
 - Lines 129-152: The structure of the manuscript goes from global forest to grass (Fig. 1), global 7 different transition types (Fig. 2) to global all vegetation change (Fig. 4). I think this could be refined to start with all vegetation transitions, broad level forest to grass and then finally explicit vegetation type transitions as this would be more logical to move towards a refinement of the detail on importance of distinguishing between the different types of vegetation transitions. At the moment this information gets lost / obscured by the priority of reporting the global all vegetation change, particularly by putting the emphasis in the abstract. Particularly after Fig. 2 it doesn’t make sense to amalgamate the SEB changes and exclude vegetation transition type when Fig. 2 demonstrates that this is actually important.
 - Line 132: “different climatic regimes” I find this hard to appreciate in Fig. 4. Perhaps split by climate region (arid/semi-arid/tropical etc.) or by SREX region. For example, I find it hard to know which pixels correspond to the boreal regions in this figure. Is 1000 to 2000 mm/yr ‘moderate rainfall’?
 - Line 139: I don’t think it works to emphasize global scale aggregates if earlier in the manuscript it is shown that regional differences exist due to different transition types.
 - Line 147: “the land cover transition during this decade has led ... to a proportional increase in surface temperature” Can the authors really attribute this to just land cover change and exclude the roll of climate change here?
 - Line 153: “the recent global signal of land cover on the surface energy budget is dominated by tropical deforestation” Fig 2 tells us that the TrBrEv to ManGra transition is the most extensive here so perhaps it would appropriate to include this here.
 - Line 158: “plant cover change” Please be consistent in terminology. Previously used “vegetation transition types”
 - Line 160: Again be consistent in terminology and use “surface energy balance”
 - Line 161: “we expect that our observation driven dataset could serve as a baseline... for the implementation of land-based climate mitigation and adaptation options” How? It is not clear enough how that link can be made with the present analysis. Particularly because the analysis considers how the vegetation transition affects local conditions but not non-local (see for example Pitman and Lorenz (2016, doi:10.1088/1748-9326/11/9/094025) and Winckler et al. (2017, doi:10.1175/JCLI-D-16-0067.1)).
- Comments on Figures Lines 165-205
1. As mentioned earlier perhaps rearrange with panels e and f at the top followed by the SEB components. Please include which time period these maps are derived from 2008-2012 or the 2000-2010.
 2. This is a very nice graphic. These are aggregated values over all grid cells where there is a particular transition type. However, if a transition type spans a large climatic range (e.g. NatGra to ManGra) could the averaging remove important information? – Particularly in the context of supplementary Figure 4 (which is not actually referred to in the narrative corresponding to Figure 2). It is also assumed that “transitions are symmetric, reverse transitions can be derived by inverting the sign” can the authors provide evidence of this in the supplementary material? Because surely there are some regions where there is for example TrBrEv to NatGra and other locations where NatGra to TrBrEv and it would be nice to see if the effect on the surface energy balance of these two different transitions are indeed symmetric.
 3. As nice as this figure is I didn’t find it added a great deal of value to the narrative. In particular if the aggregate spans diverse climatic regions, where the seasonality of temperature can vary substantially (e.g. NatGra to ManGra, Supplementary Fig. 4) I don’t think the seasonality is

apparent over such diverse climates and aggregating this would average out a lot of information. Also please note which period this corresponds to.

4. After Figures 2 and 3 this figure seems to remove all distinction on the vegetation transition types. The panel labels a/b/c/d/e/f are perhaps better to place in the bottom right corner where they don't obscure the results. Why is the contour of LE and H in Joules when the average values are reported in W/m²? It might also be more logical to show the area changed as panel a, then temperature change in b followed by the surface energy balance components to be consistent with Fig. 1.

- Line 212: "5-year epoch around the year 2010" perhaps clarify 2008-2012
 - Line 214: "parameterizations" wouldn't "empirical approaches" be more appropriate here?
 - Line 232: Perhaps rephrase to "The median value for each month is calculated from the years 2008 to 2012 to generate the 5-year climatology while retaining the seasonal cycle"
 - Line 247: Just a clarification: do the authors derive a β value for each 5x5 grid area for each PFT or a β value for each pixel and PFT that is derived from the 5x5 grid area surrounding that pixel?
 - Line 255: The Supplementary Figure 1 isn't quite a schematic it could be nice to include the β values for trees, grasses and other.
 - Line 259: Good!
 - Line 265: It is not clear to the reader here why data are aggregated to 1°.
 - Line 277: Please replace "energy fluxes. The earth's energy balance at the surface is summarized as" to "surface energy balance, which is expressed as"
 - Line 289: How are the ΔSW_{up} and ΔH values calculated per PFT transition given that the biophysical signal of vegetation change for albedo, LE and LW_{up} are based on the 5x5 0.05 moving grid area? In particular, if ΔSW_{up} and ΔH are derived at 1° how can one resolve the PFT variation in a way that is consistent with the other components of the surface energy balance?
 - Line 290: "we can safely assume that there is locally no change in incoming radiation between adjacent pixels at high resolution" I think it would be better to say "we assume no change..." unless the authors can provide references that can justify the assumptions here. Even at 0.05° resolution that is still a large area, especially considering adjacent grid cells. It disregards the effects of cloud cover.
 - Line 291: "at high resolution" I presume the authors mean the 0.05° resolution
 - Line 292: can the authors provide a justification that there are no changes in G? I understand that the idea is to reduce the number of unknowns to solve the surface energy balance but I find that the reasons given here are weak.
 - Line 294: So ΔLW_{up} and ΔLE come from 0.05° data and ΔSW_{up} from the 0.05° albedo data and the SW_{down} 1° data. Then correct ΔLW_{up} for all cloud conditions using 1° data. Then derive ΔH . How robust is this derivation to the assumptions the authors make w.r.t. SW_{down} , LW_{down} and G? It would be nice to see how well the derived quantities agree with other observational products (even if they are few). Technically, this forces the balance to be closed.
 - Line 305: Please clarify which resolution the pixels are here
 - Line 321: How are the Δy values calculated for SW_{up} and H given that they are derived from the 5x5 grid area for the other fluxes?
 - Line 322: "We aggregate these values...using CRU" I don't understand why this step is done
- Comments on Supplementary Material
- SM1 Plant Functional Type Fractions – this is rather lengthy where and could be condensed a bit by removing information that is not critical to the study
 - SM2 Preprocessing of biophysical variable datasets – here LW_{up} is actually a derived quantity by the authors using MODIS surface temperature and broadband emissivity. In the main manuscript the authors do not make this clear
 - SM4 defining locally comparable topography – it would be nice to know what proportion of pixels get discarded due to complex terrain.
 - I understand that the dataset is partially derived from 1° data and therefore can only be available at this resolution however it limits the ability to use the dataset to validate ESMs that are run a finer scale resolutions than 1° or regional climate modeling studies where the different types of vegetation transitions start to become more important for local climate
 - SM Fig. 2 Perhaps include the uncertainty on these values. Please put text in the arrows in the

same direction...makes it easier to read.

- SM Fig 3. Perhaps use different colors for LE and H, as they are difficult to distinguish.
- SM Fig 4. This figure is really handy but perhaps use a different color from green and grey to provide a better contrast.

Other comments

- Perhaps provide information on how the dataset can be accessed.

Reviewer #4 (Remarks to the Author):

This paper presents a global assessment of the effects of land cover changes on the radiative and non-radiative energy balance components of the Earth's terrestrial surface. The authors used satellite data products for their analysis to investigate in total 21 land cover transitions. They do not rely on long time series because they consider land cover variations in space rather than in time, by means of regression of the spatial cover fractions of plant function type (PFT) with energy balance components (Eq S7) in moving boxes of 25x25 km or smaller. This approach is valid because they excluded areas of steep topography.

They also consider actual land cover conversions between 2000 and 2010. These were dominated by the transitions from tropical evergreen trees into grassland and cropland. In these transitions, the upwelling longwave and shortwave radiation increase, resulting in a reduced net radiation, while the distribution of the net radiation over latent (LE) and sensible heat flux (H) and land surface temperature also change.

The study presents novel findings: (1) a conclusion on the sign of the land surface temperature change and (2) the quantification of surface energy budget changes after land cover transitions (21 in total).

These findings are of interest for a wide field, and in particular for meteorology and climatology. They provide quantitative understanding of the role of vegetation in the energy budget and Earth surface temperature, which is essential for climate modelling. The paper also clearly demonstrates the pivoting role of evaporation for land surface temperature (apart from the more direct, and much better understood effect of albedo).

One aspect that makes the study convincing is the fact that the study relies on satellite measurements rather than on modelling or limited field data.

The supplementary material is helpful and clear. The applied method only works in areas with limited topography within pixels (Suppl. 4), and the null space and spurious correlation of land cover contributions need to be removed. These steps have been carried out with appropriate methods (Suppl 3 and 4).

I have a few questions for clarification and one suggestion:

(1) How is $\text{var}(\beta)$ determined, the uncertainty in the regression coefficients (used in Eq S16)?

(2) Line 171 of Suppl 3: '... the only two points in which 2 PFT are represented have exactly the same compositions'. I presume that after the conversion of the spatial resolution from 300 m to 0.05 degrees (Line 222 of the main text), the composition values are continuous (rather than discrete as in Table S1). In that case 'exactly the same value' will in practice only occur if the value is 0 or 1 (0% or 100% cover of a PFT in a pixel). Did the cover fractions have a limited numerical precision such that mixed pixels could also have 'exactly' the same composition?

(3) I found the sentence in Line 113-115 confusing. 'These effects [i.e. the increase in outgoing

SW and LW radiation, the reduction in latent and increase in sensible heat] are stronger when forests are converted to grassland than to cropland, not least because the former [grassland] absorbed more solar energy than the latter [cropland]'. The last part of this sentence is consistent with Fig 2 (which shows that grassland reflects less and thus absorbs more solar light than cropland), but it is inconsistent with the first part of the sentence (increase in outgoing SW is stronger for the forest-> grassland conversion than for the forest -> cropland conversion).

(4) One minor point for consideration. For the λE flux, the MOD16A2 product has been used. This is a higher level satellite data driven product based on the Penman-Monteith equation. Apart from quantitative satellite data, it also uses a Biome Properties Look-Up Table (LUT) (Table 1 in Mu et al, RSE 115(8), 1781–1800) that is applied after a land cover classification to calculate the aerodynamic resistance and surface conductance that enter the Penman-Monteith equation. The satellite product makes use of some prior knowledge about the Biomes, and it is therefore not a 'pure' remote sensing product like albedo. I do not think this affects the significance of the study (for example, the Biome properties do not explain the directions of the changes in λE upon conversion for forest to grassland or cropland), but it is nevertheless worth mentioning.

C. van der Tol

Response to Reviewers for manuscript NCOMMS-17-04107A

We are glad that our research has generated a positive impression with the Editor and reviewers. We agree with the main message that most reviewers and the editors pointed out to, i.e. focusing the attention towards the effects of individual transitions. To do so, as described below, we had to change some parametrizations and redo the calculations, which we think has considerably improved the results. As requested, we have restructured to better exploit the format of *Nature Communications*, incorporated most of what was in the supplementary material. We have also drafted a sister publication in the form of a Data descriptor in *Scientific Data* to provide more details on the dataset and to make it available to the public.

Regarding the focus towards the biophysical effects of individual transitions, we had deliberately avoided it before following advice from the ESA CCI land cover team that their maps at that time (version 1.6) may not have been showing individual transitions accurately between 2000 and 2010. However, the new ESA CCI land cover maps (version 2.0.7) were released during the reviewing period of the present manuscript. Given the strong request of focusing on all vegetation transitions, we decided to reprocess the entire analysis with this new land cover dataset (which supplants the previous one according to the authors). This further ensures our generated data is based on the most up-to-date ESA maps.

The request for a focus on individual transitions also stimulated us to make sure the classes we use are as generic and understandable as possible. Also, from the comments from Reviewer 3 we feel we have put too much emphasis on the possibilities of using our data to evaluate model runs, particularly by using the logic of plant functional types (PFTs), which are essentially used by modelers. Since we had to reprocess the dataset, we decided to revisit how we defined our classes and opted for the well-known classes defined by the IGBP. By simplifying these definitions, we also found that our resulting maps had less spatial noise and less methodological uncertainty, strengthening our confidence in the general message.

Please find below our response to the specific comments of all 4 reviewers. We have set the comments of the reviewers in blue, and the answers in black.

Reviewers' comments:

Reviewer #1 (Remarks to the Author):

Duvallier et al. presents a very good synthesis of the biophysical energy response to recent land use change based solely on satellite observational changes. The analysis is solid, the paper is well written, and the focus on global energy budgets helps drive this research topic forward.

The only comment I have is that it would be useful to highlight the total energy changes by land use type (e.g. J/year globally for each of the land use classes). This could be done in a table and would further highlight the oversized role of deforestation to agricultural conversion on biophysical climatic feedbacks. If space is limiting, maybe put Figure 3 in the supplemental

We have opted in using a figure focusing on the 12 most dominant transitions to make the message clearer. It clearly highlights the oversized role of deforestation of evergreen broadleaf forests to agriculture as foreseen by the reviewer.

Reviewer #2 (Remarks to the Author):

This study presents a typical satellite-based exercise to characterise the potential or actual biophysical controls imposed by vegetation across the globe. Seems like that this work is a continuation and expansion on the Science paper [Biophysical climate impacts of recent changes in 365 global forest cover] by one of the co-authors. Overall, I think the authors did a pretty good job.

Technically speaking, the data processing and analyses appear reasonable. Given the nature of satellite measurements and many potential confounding effects and sources of uncertainty, I think it is hard to justify one approach is better than another, so here I refrain from commenting on or criticizing the technical specifics. As long as the authors acknowledge these limitations and offer caveats, I, as a reader, am happy and comfortable with the findings

From a story/implication side, I feel the authors can spend more efforts figuring out some key messages you want readers to take away; otherwise, my current reading is that they just create some information valuable to assess land cover changes--well, I appreciate this point but am left unsatisfied.

To help this point, we have changed the focus from “the production of a dataset” to “the actual changes in energy balance that recent land cover transitions have caused”, in line with the general request from reviewers 1 and 3.

No other specific comments except a minor comment on Fig.4. There, it will do readers a service if explaining what the small circles mean. Do it mean no changes or insignificant testing?

As mentioned in the caption, the small circles indicated “values that are not statistically different from zero at $p < 0.001$ ”, i.e. when the difference was less than 2 times the uncertainty represented as a standard deviation. However, we have decided to remove this figure in line with the general request of focusing on the effects of changes caused by specific land cover transitions.

Reviewer #3 (Remarks to the Author):

Manuscript Review: “The Mark of Vegetation Change of Earth’s Surface Energy

Balance”

This manuscript presents an analysis of changes in the surface energy balance as a result of different transitions in land cover. The analysis is based upon a combination of existing satellite datasets and derived products by the authors. The emphasis of the paper include: a decrease in ET when considering all vegetation change and that they use an innovative method to derive a dataset of the full surface energy balance to examine vegetation transition dependent changes in the surface energy balance. While the concept of examining the dependence of the surface energy balance to different land cover transition types is novel and a worthy contribution I feel that by focusing the main conclusions on all vegetation change the authors detract from the land use transition type dependence of their results. Furthermore, it is not clear what the main take home message is from the abstract. The manuscript reports the creation of a new global dataset of the full surface energy balance that could be used for validation of Earth System Models. There are a couple of issues with this that I think are critical for publication. See major comments for details. Generally the manuscript is well written and with further evaluation to check the robustness of the methodology of the derived surface energy balance components I do think the research would make a valuable contribution to the field if the emphasis remains on the vegetation transition types and not the aggregated values.

We appreciate these remarks and hope we have been able to address them all as mentioned below.

Major Comments

- While datasets that include the full surface energy balance (SEB) at a global scale are scarce I feel that there are some issues with how this dataset has been created.
- SWup and H are derived from the available albedo, ET and LWup and CERES datasets. Even though the albedo, ET and LWup and CERES datasets have been validated as clear in their respective publications, the two new components (SWup and H) should go a thorough validation to against other observational datasets (e.g. FLUXNET-MTE for H – see Jung et al. 2011 doi:10.1029/2010JG001566) to evaluate if the derivation method is robust to the assumptions that are made about G, SWdown and LWdown. In particular, if the idea is to propose that ESM modelers can use this dataset for validating their models, they will assume that the dataset creators have done this validation. Therefore it is critical that the authors show that their approach to derive the full SEB is valid, particularly if the intention is to share this dataset with model developers.

The SW up is the product of two "validated" datasets, namely the modis albedo dataset and the CERES radiation. The albedo should not change between clear sky and full-sky conditions, so we can safely say that the SW reflected by the surface is the SWdown from CERES times this albedo. There is therefore no need of an additional validation in our opinion.

Regarding H, we derive it as a residual term of the energy balance assuming we can disregard changes in G. We agree with the reviewer that this may not be a strong assumption (especially at monthly scale), and we have thus decided to change our claim that we are providing changes in H and instead label it as a change in the both H+G together. We are now convinced this is a more accurate label of our data and this should provide the necessary message to ESM modelers that we cannot measure H directly, only indirectly as the result of the balance of all other terms.

In this respect, this last term (H+G) depends directly from the quality of all other data sources, and we feel that making a thorough validation with source like FLUXNET-MTE are beyond the scope of this work. Such analysis may depend more on the differences in LE for instance, than on H.

Please note also that with respect to the previous version of the manuscript we have partly downplayed the objective to serve model developers. We are convinced it can be used for model developers, but that this is not the primary objective.

- I was not convinced that the assumption to disregard changes in SWdown, LWdown and G is valid. It assumes that one can ignore the effects of cloud cover over a 1° resolution despite a cloud cover 'correction' applied later to LWup. For the resolution to which SWup and H are derived, assuming zero changes in G...I don't think this can hold at that resolution. I think the authors need to provide a more compelling argument on why these assumptions are reasonable.

We have not been clear enough in our assumptions and we have now changed the text accordingly. We ignore the effects of cloud cover at the finer scale of 0.05°, not at 1°. The vegetation cover changes we are talking about are typically smaller than the 0.05° pixels. We do not pretend that this holds if we consider changes of the order of 1°. We acknowledge that if the vegetation cover change is not limited to a local change, but rather extends over a much larger coverage (i.e. more than the surface of a 0.05° pixel), changes in the cloud regime may become more prominent, and the assumption would not hold anymore. We also acknowledge that a change of 0.05° is already quite large, and that better estimates could be obtained if working at a finer resolution, but this entails much more computation time in order to make a global assessment as we have done.

Regarding the change in G, we acknowledge that the justification is weaker. It was based on the fact that annual changes might not be large, but changes at monthly scale should be more important. As proposed above, we have now decided to include changes in G in the same residual term as changes in H.

I also don't understand why the authors don't use SWdown and LWdown from the CERES EBAF-Surface dataset...these variables are included and if SWup and LWup are used so the assumption that these two terms can be ignored doesn't quite make sense if the CERES dataset has them.

Unfortunately, we cannot use SWdown and LWdown from CERES in the 'space-for-time' operation because these are only available at 1°, over which our other assumptions regarding local comparable climatic conditions would fail. And we do not use SWup and LWup directly from CERES. As explained above, the driving variable for SWup is albedo at 0.05°, we only use CERES to know how much radiation there is to reflect (the radiative effect of the land cover change is already in the change in albedo). For LWup, the driving variables are LST and emissivity at 0.05°, and CERES is again only used as a broad correcting factor to adjust from clear sky to all sky conditions (again, the info on land cover change is already included in the finer data and not in the coarser CERES pixels).

- I got the impression that the dataset covers a short time (2008-2012); I'm not sure how useful this is for ESM validation.

The metrics are derived in space, not in time, therefore the time span window is less relevant (it affects only the total number of observations). We are implicitly assuming that in a short 5 year time frame the biophysical properties are not changing, and that they remain valid to talk about the situation in 2000.

So perhaps better to avoid overselling that this is the dataset all modelers should use for validation or provide a more compelling case on why this dataset is better than LandFlux, FLUXNET-MTE, MERRA or ERA-Interim.

We have generally tuned down the emphasis that perhaps suggested that the main role of the dataset was to evaluate ESM performance. However, as discussed in the text, some properties of our dataset differentiates from those mentioned here. To retrieve the LUC signal one has to use high resolution data (5km in our case). The other datasets are at coarse resolution to explore LUC in a space for time approach. Furthermore, other products often only address one energy flux (e.g. LE for LandFlux) and not the full energy balance. FLUXNET-MTE does not report the different radiative components of the SEB (only Rn is reported)

- Analysis of the LST data between 2000 and 2010. How certain are the authors that the changes in temperature can be directly attributed to land use transitions and not climate change? I know that perhaps one can discount this due to examining differences in the SEB between vegetation types in the 5x5 grid box at the same time for the 2010 (2008-2012 epoch analysis) but I think this is missing from the manuscript.

To be precise, we are not using LST data from 2000 in our analysis. We only use the data from the 2008-2012 period to derive the local delta LST values. We consider that these delta values are valid for the (relatively) short period of time since 2000. It is correct that this is an assumption, and that a changing climate could have an effect on these. However we assume that the changes in 10 years would be relatively moderate and mitigated by considering only the local window approach. We have tried to make this clearer in the new text.

- The current structure of the manuscript could be refined by starting with the global all vegetation transition types together, global maps of the forest to grass transition and then the changes in SEB split according to the 7 transition classes. This would demonstrate how much more information is gained by refining the examination of changes in SEB as opposed to aggregating the results to all vegetation change. I think this would make the manuscript more logical.

We have changed the structure accordingly, but we start from the maps of potential change for deforestation, then go down in potential changes in different transitions, and then only into the real effects of both all vegetation changes and individual transitions. We feel the new structure streamlines the message more efficiently.

- What could be really nice is to invoke a surface energy balance decomposition with the new dataset (see examples in Luysseart et al. 2014, doi:10.1038/nclimate2196; Thiery et al. 2017, doi:10.1002/2016JD025740; or Hirsch et al. 2017, doi: 10.1002/2016JD026125) to understand which component of the surface energy balance explains the reason for temperature change for the different transition types. This could really improve the impact of the manuscript.

We have followed this suggestion and reported it in the last figure of the manuscript. The assumptions made to decompose the energy balance are reported in a new dedicated methods sub-section. The difference with respect to other studies using such decomposition is that they use it to separate direct and indirect effects, while we can only use it to separate between different direct effects (e.g. ET vs SW) as indirect effects are not assessed by our methodology. We hope that this still improves the impact of the manuscript.

- I find that the methodology description in both the main manuscript and the supplementary is thorough. There are occasions where perhaps more information or a statement justifying a particular decision /assumption would be desirable which I have noted where I feel this is necessary in the minor comments below.

- If possible, discussing the results further in the context of previous literature involving SEB dependence on the vegetation transition type would be desirable.

Minor Comments

- The title doesn't peak my interest perhaps changing to something along the lines of "Perturbation of the Surface Energy Balance Depends on Vegetation Transition Type" would be more apt.

We appreciate this concern to ensure the title is seductive to the readers. However, after giving it some thought, and having received no other indication from the 3 other reviewers to change the title, we have opted to keep the original version

- Lines 22-25: This statement refers to all vegetation change, but Line 14 alludes to the need to consider the type of vegetation change. Given what is also shown in Figure 2, it would be nice if this could be more specific to a type of vegetation change (e.g. trees to grass) rather than all vegetation change (which can include trees to grass but also grass to trees). Perhaps it would also make a stronger statement to include how much of the land area has changed over this period

We have changed the abstract inline with the redirections towards specific transitions

- Line 25: “The freely available dataset will provide valuable information to assess future scenarios of land cover change”. I’m not sure this is the best place to put such a statement because the existing narrative doesn’t really discuss how this could be done apart from as a tool to evaluate whether ESMs get the ‘right’ changes in the SEB with the different vegetation transition types...but then how can we be certain what the transitions in the future will be?

Correct. We cannot know what the future transitions will be, nor how these may be affected by concomitant climate change. We have removed this last sentence of the abstract and mentioned this point in the new discussion section.

- Line 38: Replace “in the reflectance of solar radiation” with “albedo”

Done

- Line 46: Reference 7 is also relevant here

Done

- Line 48: Replace “partition” with “partitioning”

Done

- Line 58: Replace “land’s” with “the terrestrial”

Done

- Line 59: Delete “one of”

Done

- Line 60: Replace “With the first the signal is identified” with “The first identifies the signal”

Done

- Line 63: Replace “alternative” with “second”

Done

- Line 69: Here the authors introduce the concept of “multiple vegetation transitions” and I think this could be emphasized more than it is in the abstract because this is where the research really is novel.

We agree and have tried to make it clear in the abstract.

- Line 74: “Our vegetation is defined by continuous fractions instead of crisp classes” what does this mean. Please rephrase to clarify.

We mean that we do not need to impose an arbitrary threshold to define (crisp) categorical classes. We have rephrased to clarify.

- Lines 76-79: The authors refer to using ‘mixed pixels’ and then the ‘un-mixing methodology’ ...please clarify

Basically our method can use ‘mixed pixels’ because it involves an ‘un-mixing methodology’ to get a pure signal out of these mixed pixels. We have tried to clarify by rephrasing

- Line 81: surface energy balance
Done

- Line 82: “for the epoch 2008-2012” but later on the authors compare 2000 and 2010 changes. This is a bit confusing. Furthermore if the intention is to make this data available for ESM the validation the period is short.

To avoid confusion, we have removed the years in this part, and we ensured it is clearer in the methodology. Basically, we take the horizon 2008-2012 to base the space-for-time approach and assume that it remains valid between 2000 and 2010

- Line 88: “the three class scheme” could be removed here as this is explained in the supplementary material or include a statement on what this means here.
Done

- Line 89: But also over South East Asia which generally isn’t considered a water limited area but an energy limited area (see Seneviratne et al. 2010; doi:10.1016/j.earscirev.2010.02.004)
Indeed. We have added this remark.

- Line 86-104: The narrative here starts with Figure 1e and f and then moves to Figure 1a-d. Perhaps it would be more logical to rearrange the panels of Figure 1 to match the narrative
Yes, and we have separated them in different figures since we have more flexibility with the total figure numbers

- Line 91: The references 13,14 here seem unnecessary
OK. We have removed them.

- Line 95: Include reference for “as grasses are typically brighter than trees” – even if its common knowledge
We added two references

- Line 99: “being dominated by the reduction in sensible heat in cold and/or humid climates at northern latitudes” – be careful here because over South East Asia and India there is an increase in H and large decrease in LE.
The picture has changed partly with the new calculation in which we included the transition from forest to both grasses and crops, but indeed, in parts of South Asia and South East Asia, H+G goes up and LE decreases strongly, like in Brazil. We do not imply the contrary in the text, but have tried to reformulate to be clear

- Line 105: As earlier on line 88 the authors don't define "the seven class scheme" please refer to where this is defined in the supplementary and avoid confusing the reader.

We have changed the text to clarify.

- Line 110: "decade 2000-2010" contrasting period to that reported on line 82. It's easy for the reader to get confused if two methods are used that cover different periods 2008-2012 for the 'space-for-time' logic and 2000-2010 for the other. Switching between the two here is confusing.

We have changed the text to clarify and avoid the confusion.

- Line 114: It looks like the effects of TrBrEv to ManGra is larger than TrBrEv to NatGra.

Indeed, there was an inconsistency in the old figure. With the recalculation the difference between EBF → CRO and EBF → GRA is that the conversion to GRA results in a larger decrease in ET compensated by an increase in H+G, while SW is very similar (which was not the case in the previous calculations). This simply suggests that (in the sampled areas) grasslands evapotranspire more than crops, and we have changed the text accordingly.

- Line 117: Unless I've read this wrong from Figure 2 but TrBrDe to NatGra results in an almost 10 W/m² change in LE and is larger than H. Are the authors perhaps referring to TrNeEv to NatGra here?

Yes, we we're referring to TrNeEv → NatGra. In the new calculations, there is actually a small decrease in ET now, but the comment remains valid as there is still a very strong decrease in H+G.

- Line 118: "whilst the seasonality of LE after the transition to cropland markedly follows that of the boreal summer" – I don't think this adds any value here unless the authors have a particularly point they would like to make that this transition type is mostly occurring in the Northern Hemisphere.

We have removed this.

- Line 127: "that proved the fundamental importance of biophysical processes on the long-term efficacy of land-based mitigation in Europe" – Doesn't seem fitting to have this here, the authors haven't shown this in their results and the focus of the manuscript is not on mitigation.

We wanted to stress that the conversion from needleleaf to broadleaf shows a strong effect on H, important to be considered if for mitigation options in the forestry sector, as suggested by the Naudts et al paper. We removed the sentence mentioned here by the reviewer but kept the reference.

- Lines 129-152: The structure of the manuscript goes from global forest to grass (Fig. 1), global 7 different transition types (Fig. 2) to global all vegetation change (Fig. 4). I think this could be refined to start with all vegetation transitions, broad level forest to grass and then finally explicit vegetation type transitions as this would be more logical to move towards a refinement of the detail on importance of distinguishing between the different types of vegetation transitions. At the moment this information gets lost / obscured by the priority of reporting the global all vegetation change, particularly by putting the emphasis in the abstract. Particularly after Fig. 2 it doesn't make sense to amalgamate the SEB changes and exclude vegetation transition type when Fig. 2 demonstrates that this is actually important.

We agree. We propose to start with the maps for the broad 'deforestation' transition to show the type of data we have generated, to continue with the disaggregation in different transitions (old Figure 2) and then to finish with the disaggregated effects cumulated for the 2000-2010 period. The narrative is indeed clearer in this way, now that we have the possibility to provide the latter (as mentioned in the beginning of this response to reviewers).

- Line 132: "different climatic regimes" I find this hard to appreciate in Fig. 4. Perhaps split by climate region (arid/semi-arid/tropical etc.) or by SREX region. For example, I find it hard to know which pixels correspond to the boreal regions in this figure. Is 1000 to 2000 mm/yr 'moderate rainfall'? Given the changes in the paper structure and the change in emphasis towards individual transitions (which we cannot present in this format due to the complexity), we have removed this figure.

- Line 139: I don't think it works to emphasize global scale aggregates if earlier in the manuscript it is shown that regional differences exist due to different transition types.
OK.

- Line 147: "the land cover transition during this decade has led ... to a proportional increase in surface temperature" Can the authors really attribute this to just land cover change and exclude the roll of climate change here? As mentioned before, because this is extracted from the local window space-for-time approach over which we consider unchanging climate conditions, we believe this is a valid statement.

- Line 153: "the recent global signal of land cover on the surface energy budget is dominated by tropical deforestation" Fig 2 tells us that the TrBrEv to ManGra transition is the most extensive here so perhaps it would appropriate to include this here.

We assumed that this would be understood by the term 'tropical deforestation', but we agree that we can be more specific.

- Line 158: “plant cover change” Please be consistent in terminology. Previously used “vegetation transition types”

Done

- Line 160: Again be consistent in terminology and use “surface energy balance”

Done

- Line 161: “we expect that our observation driven dataset could serve as a baseline... for the implementation of land-based climate mitigation and adaptation options” How? It is not clear enough how that link can be made with the present analysis. Particularly because the analysis considers how the vegetation transition affects local conditions but not non-local (see for example Pitman and Lorenz (2016, doi:10.1088/1748-9326/11/9/094025) and Winckler et al. (2017, doi:10.1175/JCLI-D-16-0067.1)).

We agree with the reviewer that indeed, our methodology would account only for local direct effects, and not include non-direct feedbacks, but we still think this would be a valuable start considering that local effects dominates for limited land cover transitions (as shown by Winckler et al. 2017). A more sophisticated method could use climate models to include non-local effects as well.

- Comments on Figures Lines 165-205

1. As mentioned earlier perhaps rearrange with panels e and f at the top followed by the SEB components. Please include which time period these maps are derived from 2008-2012 or the 2000-2010.

2. This is a very nice graphic. These are aggregated values over all grid cells where there is a particular transition type. However, if a transition type spans a large climatic range (e.g. NatGra to ManGra) could the averaging remove important information? – Particularly in the context of supplementary Figure 4 (which is not actually referred to in the narrative corresponding to Figure 2). It is also assumed that “transitions are symmetric, reverse transitions can be derived by inverting the sign” can the authors provide evidence of this in the supplementary material? Because surely there are some regions where there is for example TrBrEv to NatGra and other locations where NatGra to TrBrEv and it would be nice to see if the effect on the surface energy balance of these two different transitions are indeed symmetric.

Indeed, in reality there will be areas of the world experiencing change from A to B, while other areas in which B to A occurs more. However, our method relies on a static map and looks locally at neighbouring A and B classes, without ever assuming any actual change. Therefore, by construction, we obtain the same exact values for A to B than from B to A.

3. As nice as this figure is I didn't find it added a great deal of value to the narrative. In particular if the aggregate spans diverse climatic regions, where the seasonality of temperature can vary substantially (e.g. NatGra to ManGra, Supplementary Fig. 4) I don't think the seasonality is apparent over such diverse climates and aggregating this would average out a lot of information. Also please note which period this corresponds to.

The seasonality across N and S hemispheres had already been adjusted. But it is true that this might still be difficult to interpret. We have decided to move this figure, along with the old SI fig. 3 also describing the seasonality, to the *Scientific Data* data descriptor to illustrate the content of the dataset.

4. After Figures 2 and 3 this figure seems to remove all distinction on the vegetation transition types. The panel labels a/b/c/d/e/f are perhaps better to place in the bottom right corner where they don't obscure the results. Why is the contour of LE and H in Joules when the average values are reported in W/m²? It might also be more logical to show the area changed as panel a, then temperature change in b followed by the surface energy balance components to be consistent with Fig. 1.

We removed this figure in the revised version

- Line 212: "5-year epoch around the year 2010" perhaps clarify 2008-2012

- Line 214: "parameterizations" wouldn't "empirical approaches" be more appropriate here?

We consider that the weights set in the crosswalking tables are 'parameters' set by the user, so we think it is appropriate

- Line 232: Perhaps rephrase to "The median value for each month is calculated from the years 2008 to 2012 to generate the 5-year climatology while retaining the seasonal cycle"

Done

- Line 247: Just a clarification: do the authors derive a β value for each 5x5 grid area for each PFT or a β value for each pixel and PFT that is derived from the 5x5 grid area surrounding that pixel?

We use a 5x5 window that moves one pixel at a time, which to our understanding corresponds to the second option mentioned by the reviewer. And for clarification, the vector of Beta coefficients is calculated for all PFTs together.

- Line 255: The Supplementary Figure 1 isn't quite a schematic it could be nice to include the β values for trees, grasses and other.

We tried to include these extra panels in the figure, but we believe this strongly complicates the figure graphically, compromising the clarity of the message, so we decided not to add these Beta values.

- Line 259: Good!

- Line 265: It is not clear to the reader here why data are aggregated to 1°.

Indeed, this only becomes clear when closing the energy balance, as CERES data only comes at 1°. In the new version in which all the methodology was gathered together we have made this clearer.

- Line 277: Please replace "energy fluxes. The earth's energy balance at the surface is summarized as" to "surface energy balance, which is expressed as"

Done

- Line 289: How are the ΔSW_{up} and ΔH values calculated per PFT transition given that the biophysical signal of vegetation change for albedo, LE and LW_{up} are based on the 5x5 0.05 moving grid area? In particular, if ΔSW_{up} and ΔH are derived at 1° how can one resolve the PFT variation in a way that is consistent with the other components of the surface energy balance?

We confirm that the signal of the vegetation change is contained in the delta values of albedo, LE and LW_{up} obtained at the 0.05° scale. With the assumptions we make, we could try to close the surface energy balance directly at this scale to obtain Delta H at 0.05° directly. However, we would need to assume further that the CERES values for SW_{down} and the cloud correction factor ($LW_{up_C}/LW_{up_C^*}$) that are measured at 1° can be considered representative of what happens at 0.05°. Instead, we deem it more correct to bring the other 3 fluxes (albedo, LE and LW_{up}) from 0.05° to 1°, and then combining them with CERES data.

The key thing is that for the 3 fluxes containing the signal of vegetation change at 0.05°, the mean effect of this change is scaled up to 1° (taking into account possible spatial-autocorrelation as explained in the methods), but at this coarse spatial resolution we are still representing the (average) potential effect of small vegetation cover changes occurring at the scale of 0.05°, and not the effect of changing a total area of 1°x1°.

We hope these points are clearer in the revised version of the manuscript

- Line 290: “we can safely assume that there is locally no change in incoming radiation between adjacent pixels at high resolution” I think it would be better to say “we assume no change...” unless the authors can provide references that can justify the assumptions here. Even at 0.05° resolution that is still a large area, especially considering adjacent grid cells. It disregards the effects of cloud cover. We have rephrased the text to be clearer. We agree that a land cover change with a spatial extent of 0.05° is already quite large and that in some cases it could start changing cloud properties beyond what can be considered as ‘negligible’. As mentioned further above, the limitation is on the pixel size. The work could be done at finer spatial resolution to make these assumptions hold better, but the computation time to do a global assessment would increase by several orders of magnitude.

- Line 291: “at high resolution” I presume the authors mean the 0.05° resolution
Yes, we changed to be specific

- Line 292: can the authors provide a justification that there are no changes in G? I understand that the idea is to reduce the number of unknowns to solve the surface energy balance but I find that the reasons given here are weak.

We agree that the justifications are not strong, and we have thus decided to lump H and G together in the output variable resulting from the closure of the energy balance

- Line 294: So ΔLW_{up} and ΔLE come from 0.05° data and ΔSW_{up} from the 0.05° albedo data and the SW_{down} 1° data. Then correct ΔLW_{up} for all cloud conditions using 1° data. Then derive ΔH . How robust is this derivation to the assumptions the authors make w.r.t. SW_{down} , LW_{down} and G ? It would be nice to see how well the derived quantities agree with other observational products (even if they are few). Technically, this forces the balance to be closed.

Again, regarding SW_{down} and LW_{down} , we have clarified above that we assume their changes to be negligible at the finer 0.05° scale and not 1° (which is what we think the reviewer understood first). For G , we have removed the assumption as combined it with H , and the balance is closed by this joint residual term ($H+G$). We believe that these assumptions hold together well-enough for the scope of the paper and that a full comparison with other products at 1° would go beyond the scope of the study ultimately overloading an already changed document.

- Line 305: Please clarify which resolution the pixels are here 1 degree. We added it in the text.

- Line 321: How are the Δy values calculated for SW_{up} and H given that they are derived from the 5×5 grid area for the other fluxes?

ΔSW_{up} is calculated based on LST and emissivity at 0.05 (like other variables), it is then aggregated to 1 degree, and then corrected for cloud cover using 1 degree CERES data. ΔH (now $\Delta(H+G)$) is calculated from the other Δs using the equation in line 304 (of the old manuscript)

- Line 322: “We aggregate these values...using CRU” I don’t understand why this step is done

The objective was to make bins in climate space (i.e. for different sets of precipitation and temperatures), so as to make the old figure 4. Having removed that figure, we have removed this part of the manuscript.

Comments on Supplementary Material

- SM1 Plant Functional Type Fractions – this is rather lengthy where and could be condensed a bit by removing information that is not critical to the study

We have adapted (and simplified) this part given that we now use IGBP classes instead of PFTs.

- SM2 Preprocessing of biophysical variable datasets – here LW_{up} is actually a derived quantity by the authors using MODIS surface temperature and broadband emissivity. In the main manuscript the authors do not make this clear

We think that now, in the revised version that includes all the methodology together, this should be clearer.

- SM4 defining locally comparable topography – it would be nice to know what proportion of pixels get discarded due to complex terrain.

About 20% of emerged surfaces are discarded due to complex terrain. We have been rather conservative with our thresholds, tolerating little change in topography to ensure this does not influence our results.

- I understand that the dataset is partially derived from 1° data and therefore can only be available at this resolution however it limits the ability to use the dataset to validate ESMs that are run a finer scale resolutions than 1° or regional climate modeling studies where the different types of vegetation transitions start to become more important for local climate

Yes, this is a limitation that we cannot fully escape given the spatial resolution of CERES. However, it should be noted that even if the spatial support of this dataset is 1 degree, it is representative of changes at a much finer scale (0.05 degree), and that is true even for H+G since it is derived as a difference of all other terms which all come from 0.05 pixels.

- SM Fig. 2 Perhaps include the uncertainty on these values. Please put text in the arrows in the same direction...makes it easier to read.

We placed the values in the same direction. However, we did not have actual uncertainties as we do not have an estimation of the error of each measure. What we added instead are the mean of the 12 monthly standard deviations of all values across the world. Therefore the value indicates more the spatial variability than the uncertainty.

- SM Fig 3. Perhaps use different colors for LE and H, as they are difficult to distinguish.

In the current version, we have removed the figure (it will go in the data descriptor, and we will consider changing the colours)

- SM Fig 4. This figure is really handy but perhaps use a different color from green and grey to provide a better contrast.

OK, we changed it to orange, and included it in the main text.

Other comments

- Perhaps provide information on how the dataset can be accessed.

We have the intention to distribute it in FigShare along with a 'data descriptor' in Scientific Data. We hope that will be ready by the time the present paper is published so that the references are correct.

Reviewer #4 (Remarks to the Author):

This paper presents a global assessment of the effects of land cover changes on the radiative and non-radiative energy balance components of the Earth's terrestrial surface. The authors used satellite data products for their analysis to investigate in total 21 land cover transitions. They do not rely on long time series because they consider land cover variations in space rather than in time, by means of regression of the spatial cover fractions of plant function type (PFT) with energy balance components (Eq S7) in moving boxes of 25x25 km or smaller. This approach is valid because they excluded areas of steep topography.

They also consider actual land cover conversions between 2000 and 2010. These were dominated by the transitions from tropical evergreen trees into grassland and cropland. In these transitions, the upwelling longwave and shortwave radiation increase, resulting in a reduced net radiation, while the distribution of the net radiation over latent (LE) and sensible heat flux (H) and land surface temperature also change.

The study presents novel findings: (1) a conclusion on the sign of the land surface temperature change and (2) the quantification of surface energy budget changes after land cover transitions (21 in total).

These findings are of interest for a wide field, and in particular for meteorology and climatology. They provide quantitative understanding of the role of vegetation in the energy budget and Earth surface temperature, which is essential for climate modelling. The paper also clearly demonstrates the pivoting role of evaporation for land surface temperature (apart from the more direct, and much better understood effect of albedo).

One aspect that makes the study convincing is the fact that the study relies on satellite measurements rather than on modelling or limited field data.

The supplementary material is helpful and clear. The applied method only works in areas with limited topography within pixels (Suppl. 4), and the null space and spurious correlation of land cover contributions need to be removed. These steps have been carried out with appropriate methods (Suppl 3 and 4).

I have a few questions for clarification and one suggestion:

(1) How is $\text{var}(\beta)$ determined, the uncertainty in the regression coefficients (used in Eq S16)?

Beta is obtained directly from the regression (eq 8 in the new version), and $\text{var}(\beta)$ is the variance-covariance matrix of Beta calculated in R using the 'vcov' function in the 'stats' base package.

(2) Line 171 of Suppl 3: '... the only two points in which 2 PFT are represented have exactly the same compositions'. I presume that after the conversion of the spatial resolution from 300 m to 0.05 degrees (Line 222 of the main text), the composition values are continuous (rather than discrete as in Table S1). In that

case 'exactly the same value' will in practice only occur if the value is 0 or 1 (0% or 100% cover of a PFT in a pixel). Did the cover fractions have a limited numerical precision such that mixed pixels could also have 'exactly' the same composition?

We did limit the numerical precision of the fractions to 4 decimal places. But also, although the conversion goes from discrete to continuous, if in two distinct 0.05 degree pixels there are exactly the same proportions of the same classes at 300m (i.e. half forest, half grass), we may end up in the same numerical values even for other cases than 0 or 100%. It doesn't happen often, but sometimes it does.

(3) I found the sentence in Line 113-115 confusing. 'These effects [i.e. the increase in outgoing SW and LW radiation, the reduction in latent and increase in sensible heat] are stronger when forests are converted to grassland than to cropland, not least because the former [grassland] absorbed more solar energy than the latter [cropland]'. The last part of this sentence is consistent with Fig 2 (which shows that grassland reflects less and thus absorbs more solar light than cropland), but it is inconsistent with the first part of the sentence (increase in outgoing SW is stronger for the forest-> grassland conversion than for the forest-> cropland conversion).

We have rephrased this part to make things clearer.

(4) One minor point for consideration. For the λE flux, the MOD16A2 product has been used. This is a higher level satellite data driven product based on the Penman-Monteith equation. Apart from quantitative satellite data, it also uses a Biome Properties Look-Up Table (LUT) (Table 1 in Mu et al, RSE 115(8), 1781-1800) that is applied after a land cover classification to calculate the aerodynamic resistance and surface conductance that enter the Penman-Monteith equation. The satellite product makes use of some prior knowledge about the Biomes, and it is therefore not a 'pure' remote sensing product like albedo. I do not think this affects the significance of the study (for example, the Biome properties do not explain the directions of the changes in λE upon conversion for forest to grassland or cropland), but it is nevertheless worth mentioning.

We have mentioned this in the description of the LE data in the methods section.

C. van der Tol

Reviewers' comments:

Reviewer #1 (Remarks to the Author):

The authors have made the suggested revisions that I put forth in my previous review of an earlier version of this manuscript. I have no further suggested revisions.

Reviewer #3 (Remarks to the Author):

Review of the revised manuscript "The mark of vegetation change on Earth's surface energy balance"

Summary:

I welcome the efforts in restructuring the manuscript. It is now much easier to follow the narrative of the manuscript that the authors have substantially clarified and simplified in most, but not all cases.

I appreciate that the authors have reduced the emphasis on creating a new dataset, where this will be published elsewhere. However, in the process of this restructuring it is still not clear to me what the main take home message of this manuscript is, particularly after reading the Discussion section and abstract where there is no high impact statement one would expect in a Nature paper. While the analysis is novel I don't know what the main result is apart from "we show how changes in SEB vary with vegetation transition type". This could be improved with stronger wording. They claim that "we then use this data to quantify the impact of actual vegetation changes..." impact on what? On extremes? On heatwaves? It is not enough just to say that there is a change in ET and albedo from deforestation. This is already known from the wealth of literature that has examined the impacts of historical climate change on climate. This makes me think that the key result is the derivation of the dataset rather than what the transition type analysis shows. Furthermore the methodology is long (19 pages vs. 9 for the main text) which makes me feel that this study would be more relevant for publication in another journal such as Remote Sensing and Environment. To have more impact, it might be better to say something like: "Recent tropical deforestation has contributed to X impacts on Y that can be explained by changes in the SEB derived using the first global scale observational dataset of the full SEB."

In its current form, if the methodology and dataset produced by the methodology are to be published elsewhere, as implied by the response to reviewer comments, then what does this paper actually achieve?

Major Comments:

1. I get that for a global dataset there are temporal constraints in the remote sensing data, but this 'space-for-time' approach makes me uneasy. I think you need to provide a strong statement to justify this approach, particularly because the datasets used do have some temporal coverage: ESA CCI landcover has 3 epochs including 2000 and 2010; albedo is 2000 to present; LHF is 2000 to 2010 and LST starts in 2002...although not so clear on the website if this goes to present day or some time earlier. If you have data that spans time, I don't understand why one would not consider changes between 2010 and 2002? Perhaps I misunderstood this but if you have temporal data then I would conduct the analysis using this approach rather than a 'space-for-time' approach. Certainly on lines 201-203 I get the impression that such a temporal analysis was done for parts of the analysis...which makes this confusing particularly because this seems to contradict then the narrative at the beginning of the manuscript.

2. Lines 84-86: "Instead, we use the actual cover fraction value for different vegetation types

within a mixed pixel as predictors, thereby unmixing the confounding factors of each vegetation type.” – In the first review I asked for clarification on what the ‘unmixing methodology’ means. The response from the authors did not resolve this. I still do not understand what you mean by unmixing confounding factors or what an unmixing approach means. This reads as jargon particular to remote sensing that may not be self explanatory to all readers.

3. I was disappointed that my request to validate the robustness of the methodology used to derive the dataset was dismissed so easily. If the authors plan to publish the dataset elsewhere, it is likely that a reviewer will ask for this. In particular, if the argument for not doing the validation is “we already use validated datasets to create this new dataset” (the authors words not mine!) then the narrative needs to be reframed. I have a preference to use longer datasets that sample interannual variability for ESM validation. If the authors are lumping H and G together as a residual term from R_n and LE then I can’t see how this ‘new dataset’ adds value given the short temporal coverage if I can derive a similar H+G residual from gridded products such as GLEAM, LandFlux-EVAL, Fluxnet-MTE and GEWEX-SRB. Here the authors need to emphasise that the difference is that the SEB has been split into the different vegetation transitions that these other products cannot do. Perhaps making this clear would assist in communicating the impact of the study.

4. The results now follow a more logical structure, which is an improvement from the initial submission. However in the discussion, statements such as the following “The dataset we have generated is currently based on products developed for various applications at global scale, such as the evaluation of climate impacts of future scenarios of vegetation cover change within the framework of integrated assessment models. At regional scale the method could be eventually applied with higher-resolution inputs such as more accurate biophysical variables and detailed thematic maps (e.g. areas with different land management practices).” Requires the reader to take a leap of logic. Firstly the authors examine present day vegetation transition dependence on the change in LST via SEB changes. The biophysical feedbacks of LUC are important but then clearly articulate what they are and then relate to the implications for future mitigation of climate change. Furthermore, it is first in the Discussion where the methodology is promoted as the key product of this manuscript. This needs to be reframed and introduced earlier. The results provide useful insights into how the biophysical feedbacks of different vegetation transitions could be exploited for mitigation but this is not clearly articulated in the manuscript. Similarly, I’m not so sure it makes sense to bring in commentary about land geoengineering here...its not so relevant to the purpose of the study.

5. The data availability declaration is rather vague and needs to be resolved prior to publication either with a doi of the published dataset or contact details.

Minor Comments:

Line 90: Perhaps if you use the epoch 2008-2012 it doesn’t quite make sense to talk about vegetation change during 2000-2010.

Line 143: remove ‘destination’ – not necessary

Line 145-147: “suggesting grasslands have a longer growing season or more access to water in the analysed areas” – this is something you presume. Could this be related to the fractional cover...i.e. there is a larger area converted to grass than crop?

Line 153-164: This paragraph doesn’t really add value apart from ‘advertising’ the dataset. Perhaps move to methods...

Line 237-243: “Being able to tackle directly this local effect is an advantage that observation-

driven assessments have over model-based ones, which either have problems disentangling the low land cover change signal from climate noise in their large pixels[19, 32], and so have resort to large scale idealized simulations in which local and non-local effects are intermingled[12, 33], or have to develop extra methodologies to isolate local effects[23, 34].” – Be careful here. With an ESM you can run offline simulations to suppress spatial feedbacks and therefore the non-local impact. Then if you compare this then to coupled (land-atmosphere) simulations you can evaluate the effect of the two-way interaction. This is often why models are used because you can isolate parts of the climate system to understand particular feedbacks which you can't do through observations because everything is there. Studies such as GLACE (e.g. Koster et al. 2004 Science) are a good example of this.

“Preprocessing of biophysical variable datasets” section – can shorten the definitions of albedo, LE to reduce length.

Compared to the initial submission, the entire methodology has been moved from the supplementary material to the main paper. This is awfully long and I don't think the length now adheres to the limits for Nature Comms “Methods are typically less than 3000 words.” I get the impression from this that the study is more a methods paper with an example application. Without clear high impact statements which are currently lacking in the manuscript it is perhaps more suitable for publication in another journal rather than in Nature.

I still noted several instances where sentences are missing words. I have not noted them here but would expect all authors to carefully check the wording of the manuscript.

Comments on Figures:

Figure 1 doesn't really add value if the authors don't quote the SEB values in the manuscript text. Figure 1 caption: replace “the balance of the other fluxes” with “the residual of the other fluxes”

Figure 5/6 – perhaps unbold the values in the panels, because they are hard to read when you view the figure on an A4 page.

Figure 6 – I think the panel on the right is the only one that is necessary.

Reviewer #4 (Remarks to the Author):

The authors have addressed all points in my first review. I have no further comments on the revised manuscript.

Response to reviewers (second round)

We have revised the manuscript. Please find below the reviewers and editor's remarks in blue italics, followed by our responses below in black.

Reviewer #3 (Remarks to the Author):

Review of the revised manuscript "The mark of vegetation change on Earth's surface energy balance"

Summary:

I welcome the efforts in restructuring the manuscript. It is now much easier to follow the narrative of the manuscript that the authors have substantially clarified and simplified in most, but not all cases.

I appreciate that the authors have reduced the emphasis on creating a new dataset, where this will be published elsewhere. However, in the process of this restructuring it is still not clear to me what the main take home message of this manuscript is, particularly after reading the Discussion section and abstract where there is no high impact statement one would expect in a Nature paper. While the analysis is novel I don't know what the main result is apart from "we show how changes in SEB vary with vegetation transition type". This could be improved with stronger wording. They claim that "we then use this data to quantify the impact of actual vegetation changes..." impact on what? On extremes? On heatwaves? It is not enough just to say that there is a change in ET and albedo from deforestation. This is already known from the wealth of literature that has examined the impacts of historical climate

change on climate. This makes me think that the key result is the derivation of the dataset rather than what the transition type analysis shows. Furthermore the methodology is long (19 pages vs. 9 for the main text) which makes me feel that this study would be more relevant for publication in another journal such as Remote Sensing and Environment. To have more impact, it might be better to say something like: "Recent tropical deforestation has contributed to X impacts on Y that can be explained by changes in the SEB derived using the first global scale observational dataset of the full SEB." In its current form, if the methodology and dataset produced by the methodology are to be published elsewhere, as implied by the response to reviewer comments, then what does this paper actually achieve?

We have worked on articulating a more impactful message. We now stress how this is the first data-driven assessment of the potential effect on the full surface energy balance of multiple vegetation transitions at global scale. We have consolidated the main outcome as having quantified an increase of 0.23 K in local land surface temperature as a result of the total perturbation of the SEB caused by all vegetation changes over the period 2000-2015. We have identified that agricultural expansion at the expense of EBF, SHR and DBF are the main vegetation transitions responsible for this change; and we are capable to identify and quantify the mechanisms behind this effect (decrease in LE which is larger than the increase in albedo). Note that we have extended the period of analysis from 2010 to 2015 (the latest year for which we have land cover maps) in order to be more up-to-date. Throughout the text, we have also tried to further minimize the focus on the dataset, leaving that for the data descriptor paper, but maintaining here the originality of having a unique global assessment based on that dataset.

Concerning the dataset, we submitted it to a data paper journal as suggested by the editor to ensure that the data be accessible. That specific data paper requires information on the methodology for reasons of completion, but for them the novelty of the methodology remains with the original paper (i.e. this manuscript in Nature Communications). Therefore, we consider that the methodology remains an additional novelty to this work (beyond the full-blown assessment as described above).

Major Comments:

1. I get that for a global dataset there are temporal constraints in the remote sensing data, but this 'space-for-time' approach makes me uneasy. I think you need to provide a strong statement to justify this approach, particularly because the datasets used do have some temporal coverage: ESA CCI landcover has 3 epochs including 2000 and 2010; albedo is 2000 to present; LHF is 2000 to 2010 and LST starts in 2002...although not so clear on the website if this goes to present day or some time earlier. If you have data that spans time, I don't understand why one would not consider changes between 2010 and 2002? Perhaps I misunderstood this but if you have temporal data then I would conduct the analysis using this approach rather than a 'space-for-time' approach. Certainly on lines 201-203 I get the impression that

such a temporal analysis was done for parts of the analysis...which makes this confusing particularly because this seems to contradict then the narrative at the beginning of the manuscript.

Data from MODIS is indeed available since 2000. However, we are not using the space-for-time approach to avoid using that data. If we were to use the changes in time from 2000 onwards we would be limited in our analysis to the areas where land cover changes have occurred in the last 15 years. While deforestation events are rather common and (relatively) easy to detect, other transitions occur much less frequently in space and time (e.g. reforestation, which is more gradual) or are difficult to detect (e.g. from grasses to crops). In some regions (e.g. Europe) there has been little changes in land cover in the past 15 years, so limiting our study to the areas with recent change in land cover would result in very limited observations restricted to some World regions of few land cover transitions. This assessment would have therefore been incomplete both in term regional cover and land cover types.

By looking at the different land cover situations over a local vicinity, the space-for-time approach gives us two main advantages: (1) it allows us to explore transitions that have not yet occurred and (2) it allows us to have much larger spatial coverage, since we can estimate the effect of a potential change between two plant cover type wherever they coexist. Once we obtain the local variation in each biophysical variable (ET, LST, albedo) for every considered vegetation transition and at every location, we can use these estimates to reconstruct the recent history. Our assumption is that the mean signal of a give transition calculated on the 2008-2012 period is representative of what would happen (on average) at any moment during the larger time span of the analysis (2000-2015). In other words, we consider that if we calculate for the time window 2008-2012 that a full transition from trees to grasses leads to an average change of +1 K in LST in a given place, this value would also apply for a similar full transition from trees to grasses happening from 2000 to 2015 over the same place. For this biophysical variables this approach should hold irrespective of inter-annual variability since we are looking at average values (2008-2012) and extrapolate the mean signal only for few years in the past and in the future. On a 15 years temporal interval we don't expect to see any trend in the biophysical signal and therefore we assume that the variability is a random signal that is properly smoothed by the temporal averaging in the 2008-2012 time window.

To move beyond and grasp the inter-annual variability and see the impact of extremes, we could apply this space-for-time approach on every year from 2000 till 2015 using each individual (yearly) land cover maps from the ESA CCI along with the corresponding data from MODIS. However, this would reduce substantially the number of observations and therefore the estimates would be more uncertain. Exploring the interannual variability of the biophysical land signal therefore goes beyond the scope of the current work.

2. Lines 84-86: "Instead, we use the actual cover fraction value for different vegetation types within a mixed pixel as predictors, thereby unmixing the

confounding factors of each vegetation type.” – In the first review I asked for clarification on what the ‘unmixing methodology’ means. The response from the authors did not resolve this. I still do not understand what you mean by unmixing confounding factors or what an unmixing approach means. This reads as jargon particular to remote sensing that may not be self explanatory to all readers.

We regret that the first revision did not succeed in clarifying this point. To improve the clarity and avoid jargon, in the revised version of the main text we have removed the reference to 'unmixing'. Please note that we do not expect the reader to understand all the technical details of this approach only from this introduction, but rather from reading the detailed methodological section. Basically, the pixels we are working with (at 5km of spatial resolution) very often have mixed land cover types, which will have different effects on the land surface properties recorded in the pixel. When we ‘un-mix’, we refer to the process of disentangling the relative contributions of each vegetation fraction within the pixel to the overall variable (LST, albedo, LE) that we are interested in. This is what we do with the local regression in which we use the vegetation fractions as explanatory variables and the biophysical variable (LST, albedo, LE) as response variable.

3. I was a disappointed that my request to validate the robustness of the methodology used to derive the dataset was dismissed so easily. If the authors plan to publish the dataset elsewhere, it is likely that a reviewer will ask for this. In particular, if the argument for not doing the validation is “we already use validated datasets to create this new dataset” (the authors words not mine!) then the narrative needs to be reframed. I have a preference to use longer datasets that sample interannual variability for ESM validation. If the authors are lumping H and G together as a residual term from Rn and LE then I can’t see how this ‘new dataset’ adds value given the short temporal coverage if I can derive a similar H+G residual from gridded products such as GLEAM, LandFlux-EVAL, Fluxnet-MTE and GEWEX-SRB. Here the authors need to emphasise that the difference is that the SEB has been split into the different vegetation transitions that these other products cannot do. Perhaps making this clear would assist in communicating the impact of the study.

This is correct, our methodology based on satellite data at 0.05 decimal degree is capable of splitting the SEB into different vegetation transitions and this is not feasible with the products mentioned above due to their excessively coarse spatial resolution. This is a fundamental difference between our dataset and these other products. A proper comparison with these products would be to apply the methodology we have developed on these datasets to then obtain the (un-mixed) effects of each vegetation transition. However, applying the method to these datasets is open to strong criticism because of the much coarser pixels that they have. The local window over which we would need to un-mix would be too large (~140 x 140 km at the equator) to assume that the landscape and climate conditions are sufficiently homogeneous within the spatial domain of the regression.

Nevertheless, to satisfy this concern for validation, we have now provided an extra figure in the supplementary material in which we do a comparison of the original

MODIS LE product (arguably the one that is most open to criticism) with the coarsened resolution GLEAM LE, which is currently considered a better option.

Given that our methodology for the "unmixing" is not applicable to GLEAM (for its coarse resolution) the analysis is done by comparing all pixels with Forest Cover > 75% and all pixels having Forest Cover < 25%. The global patterns of the LE signal in the forest - non forest transition are very comparable between MODIS and GLEAM, which builds confidence on the fact that MODIS can be used in our analysis.

4. The results now follow a more logical structure, which is an improvement from the initial submission. However in the discussion, statements such as the following "The dataset we have generated is currently based on products developed for various applications at global scale, such as the evaluation of climate impacts of future scenarios of vegetation cover change within the framework of integrated assessment models. At regional scale the method could be eventually applied with higher-resolution inputs such as more accurate biophysical variables and detailed thematic maps (e.g. areas with different land management practices)." Requires the reader to take a leap of logic.

With these phrases, we wanted to highlight that the methodology can be adapted to a finer scale when targeting regional applications by ingesting data at finer spatial resolution and with more thematic detail. Following the suggestion of the reviewer we have rephrased this section to be clearer.

Firstly the authors examine present day vegetation transition dependence on the change in LST via SEB changes.

It is the other way around. We examine the changes in LST that could occur via SEB changes caused by potential vegetation transitions.

The biophysical feedbacks of LUC are important but then clearly articulate what they are and then relate to the implications for future mitigation of climate change. Furthermore, it is first in the Discussion where the methodology is promoted as the key product of this manuscript. This needs to be reframed and introduced earlier.

Following the suggestion of the reviewer we have adapted the manuscript to ensure that the main message is the quantification of how much perturbation LULCC has had in the 2000-2015 period. The methodology remains an important secondary product of this manuscript.

The results provide useful insights into how the biophysical feedbacks of different vegetation transitions could be exploited for mitigation but this is not clearly articulated in the manuscript.

We have now restructured the discussion to begin a paragraph explicitly articulating this aspect.

Similarly, I'm not so sure it makes sense to bring in commentary about land geoengineering here...its not so relevant to the purpose of the study.

We agree. The term has been removed.

5. The data availability declaration is rather vague and needs to be resolved prior to publication either with a doi of the published dataset or contact details.

This was vague because at the time of writing we had not yet received any feedback from the data descriptor journal. Now that we have the green light to proceed from their side we can be more explicit.

Minor Comments:

Line 90: Perhaps if you use the epoch 2008-2012 it doesn't quite make sense to talk about vegetation change during 2000-2010.

The manuscript is using two different time windows because it builds on a two steps procedure:

- 1) The median values around the period 2008-2012 are used in the 'space-for-time' exercise to obtain the median 'delta' for a given variable, at a given place and for a given transition.
- 2) This 'delta' observed a is assumed to be valid for similar transitions over the same area over the period 2000 onwards, and therefore in a second step we use the deltas to estimate how much change has occurred from 2000 until (in this revised version) 2015.

Given the short time period of the analysis we believe that this methodology may deliver robust and consistent results.

Line 143: remove 'destination' – not necessary

Done

Line 145-147: "suggesting grasslands have a longer growing season or more access to water in the analysed areas" – this is something you presume. Could this be related to the fractional cover...i.e. there is a larger area converted to grass than crop?

It shouldn't be the case since the methodology extracts this value independently of fraction area. The result of each local regression provides an estimate of how a pixel fully covered with forest would react when converted fully to grasslands or to croplands. Thanks to our "unmixing" method we can derive metrics that are independent on the fractional cover, even if there are no fully forested or fully grassland/croplands pixels.

Line 153-164: This paragraph doesn't really add value apart from 'advertising' the dataset. Perhaps move to methods...

We have removed this paragraph from the main text

Line 237-243: "Being able to tackle directly this local effect is an advantage that observation-driven assessments have over model-based ones, which either have problems disentangling the low land cover change signal from climate noise in their large pixels[19, 32], and so have resort to large scale idealized simulations in which local and non-local effects are intermingled[12, 33], or have to develop extra methodologies to isolate local effects[23, 34]." – Be careful here. With an ESM you can run offline simulations to suppress spatial feedbacks and therefore the non-local impact. Then if you compare this then to coupled (land-atmosphere) simulations you can evaluate the effect of the two-way interaction. This is often why models are used because you can isolate parts of the climate system to understand particular feedbacks which you can't do through observations because everything is there. Studies such as GLACE (e.g. Koster et al. 2004 Science) are a good example of this.

We agree with the reviewer that indeed models have the advantage that different parts of the climate system can be studied in isolation within their simulations. However, as we say, they are currently limited by the coarse spatial resolution of their grids cells, which is arguably too coarse to properly disentangle the local effects of various PFTs. Perhaps this could be done by perturbing some pixels of regional climate models that are nested in ESMs, but the pixels would still need to be very small (<5km) for modelling standards to reach the same level of detail that we are doing here.

Regarding non-local effects, we have explicitly added a caveat that these are not targeted by this work, which focuses only on the local effects:

"It is worth noting that our approach addresses exclusively the direct biophysical impact of land cover change at local scale, since the climate feedback and large scale teleconnections cannot be assessed with the proposed methodology of local space-for-time substitution.

"Preprocessing of biophysical variable datasets" section – can shorten the definitions of albedo, LE to reduce length.

We agree with the suggestion and restrict the more detailed description on the dataset paper.

Compared to the initial submission, the entire methodology has been moved from the supplementary material to the main paper. This is awfully long and I don't think the length now adheres to the limits for Nature Comms "Methods are typically less than 3000 words."

We have followed the guidelines of the editor in order to comply with the specific requirements of Nature Communication. Here are the instructions we received from the editor:

I would also like to take this opportunity to draw your attention to the more generous content allowance offered by Nature Communications in comparison to the Nature research journals. Nature Communications permits articles to be up to 5000 words in length, plus an unlimited Methods section

I get the impression from this that the study is more a methods paper with an example application. Without clear high impact statements which are currently lacking in the manuscript it is perhaps more suitable for publication in another journal rather than in Nature.

Following the advice of the reviewer in the new version of the manuscript we have focused on a high-level outcome of the study, namely the global cumulative effect of all land cover transition observed in the last 15 years on the surface energy balance and local temperature.

We believe that this is an high impact result, that is i) purely based on Earth observation of energy fluxes, ii) relies on the most updated description of recent changes in global land cover and iii) is based on a novel methodology that overcome the limits of previous studies in the field and offer a robust estimates of the uncertainties.

I still noted several instances where sentences are missing words. I have not noted them here but would expect all authors to carefully check the wording of the manuscript.

We have done so.

Comments on Figures:

Figure 1 doesn't really add value if the authors don't quote the SEB values in the manuscript text. Figure 1 caption: replace "the balance of the other fluxes" with "the residual of the other fluxes"

We have removed the figure

Figure 5/6 – perhaps unbold the values in the panels, because they are hard to read when you view the figure on an A4 page.

Thank you for the suggestion. We will adjust these cosmetic details once we prepare the final version of the figures in case of acceptance of the paper.

Figure 6 – I think the panel on the right is the only one that is necessary.

We have retained the right panel, and placed the left one in supplementary material

Reviewer #5 (Remarks to the Author):

I think this is an interesting paper, that is certainly a useful approach and so far as I can tell its been done well. However, I was asked to largely focus on the second round review and the responses by the authors to those reviews.

I am afraid I largely agree with the reviewer. I think the key issue is highlighted right up front in their review. The reviewer states:

"However, in the process of this restructuring it is still not clear to me what the main take home message of this manuscript is, particularly after reading the Discussion section and abstract where there is no high impact statement one would expect in a Nature paper. While the analysis is novel I don't know what the main result is apart from "we show how changes in SEB vary with vegetation transition type".

I agree. The paper is good, it is interesting and it is worth publishing but I do not see the key message either. I do not see why I need to read this paper - I most certainly would want to explore the data and likely use it, but the paper itself just misses the key impact statements to make this paper Nature Communications worthy.

I am not sure why the authors have not taken this reviewer's comments more on board. Perhaps they feel they are, but I do not see it. I think the review is fair and quite a nice roadmap for the authors - that is the reviewer is trying hard to help the authors make this Nature Communications worthy. But rather than take this road I think the authors have not quite grasped the difference between good technical information and a clear and open statement of intent and impact.

Final modifications done to manuscript NCOMMS-17-04107C

We have proceeded to make the changes necessary to ensure the main messages are expressed with stronger statements, as requested by the reviewers and the editor. We have made all the changes requested by the editor in the commented PDF document, as it can be seen in the revised manuscript with annotated changes.

These changes consist essentially in:

- rewriting the last paragraph of the introduction to remove some more technical insights of the method and focusing on the main outcomes of the study
- adding some further description of the patterns in figure 3 (previously figure 5)
- reorganizing the discussion to strengthen the message, to discuss the value of the methodology and to focus more on the message on figure 3 (previously figure 5) regarding the implication for land-based climate mitigation

In order to further separate the analysis provided in this paper from the dataset in described in the associated publication in *Scientific Data*, we have been asked by the *Scientific Data* editor to remove from that paper two figures showing the temporal variations of changes in SEB and T per vegetation cover class transition. Since these add complementary value to the *Nature Communications* figure 3 (previously figure 5), which has been suggested by the editor as being a key figure, we have added them in the supplementary information of *Nature Communications*.

Since the last revision of this paper, we also made a minor revision of the dataset that has resulting in small numerical changes. These changes have been propagated to the revised figures, which can show minor modifications since the previous version, but they do not change in any way the results nor the conclusions. We wanted to make these changes now to ensure the paper is based on the latest version of the data.

We have gone through the entire manuscript to ensure we follow the guidelines in terms of format for the text, the mathematical notation and the figures (i.e. ensuring colourblind palettes are used where appropriate).

In the supplementary information, we have added titles and ensured all comments have been addressed, following the guidelines in the webpage.